



# Analogue modelling of basin inversion: a review and future perspectives

Frank Zwaan[1], Guido Schreurs[1], Susanne J.H. Buiter[2], Oriol Ferrer[3], Riccardo Reitano[4], Michael Rudolf[5,6], Ernst Willingshofer[7]

[1] University of Bern, Institute of Geological Sciences, Bern, Switzerland
[2] RWTH Aachen University, Faculty of Georesources and Materials Engineering, Tectonics and Geodynamics, Aachen, Germany
[3] Universitat de Barcelona, Facultat de Ciències de la Terra, Departament de Dinàmica de la Terra i de l'Oceà, Geomodels Research Institute, Barcelona, Spain
[4] Università Degli Studi Roma Tre, Department of Geological Sciences, Rome, Italy
[5] Helmholtz Centre Potsdam GFZ German Research Centre for Geosciences, Potsdam, Germany
[6] Technical University Darmstadt, Institute for Applied Geosciences - Engineering Geology, Darmstadt, Germany
[7] Utrecht University, Department of Earth Sciences, Utrecht, The Netherlands

*Correspondence to*: Frank Zwaan (frank.zwaan@geo.unibe.ch)

**Abstract.**

Basin inversion involves the reversal of subsidence in a basin due to compressional tectonic forces, leading to uplift of the basin's sedimentary infill. A thorough understanding of basin inversion is of great importance for scientific, societal and economic reasons. Analogue tectonic modelling forms a key part our efforts to improve our understanding of basin inversion processes, and researchers have conducted numerous studies on this topic. In this review paper we recap the advances in knowledge of basin inversion tectonics acquired through analogue modelling studies, providing an up-to-date summary of the state of analogue modelling of basin inversion. We describe the different definitions of basin inversion that are being applied by researchers, why basin inversion has been historically an important research topic, and what the general mechanics involved in basin inversion are. We subsequently treat the wide range of different experimental approaches used for basin inversion modelling, with attention to the various materials, set-ups and techniques used for monitoring and analysing the model results. Our new systematic overviews of generalized results reveal the diversity of model results, depending greatly on the chosen set-up, model layering and (oblique) kinematics of inversion, as well as 3D along-strike structural and kinematic variations in the system. We show how analogue modelling results are in good agreement with numerical models, and how these results help to better understand natural examples of basin inversion. In addition to reviewing the past efforts in the field of analogue modelling, we also shed light on future modelling challenges and identify a number of opportunities for follow-up research. These include the testing of force-boundary conditions, adding geological processes such as sedimentation, transport and erosion, applying state-of-the-art modelling and quantification techniques, and establishing best modelling practices. We also suggest expanding the scope of basin inversion modelling beyond the traditional upper crustal "North Sea style" of inversion, which may contribute to the on-going search for clean energy





resources. It follows that basin inversion modelling can bring valuable new insights, providing a great incentive to continue our efforts in this field. We therefore hope that this review paper will form an inspiration for future analogue modelling studies of basin inversion.

## 1. Introduction

### 1.1. Definition of basin inversion

The development of extensional tectonic systems leads to the formation of (fault-bounded) basins, followed by crustal necking and eventually continental break-up and oceanic spreading (e.g. Lavier & Manatschal 2006; Wilson et al. 2019). However, at any time during continental break-up, changes in the tectonic regime may halt rifting and lead to subsequent compression, causing the inversion of the previously established basins (Fig. 1).

Even though the concept of basin inversion has been used since over a century ago (e.g. Lamplugh 1919), the term "inversion" appears to have been initially introduced by Glennie & Boegner (1981), who used *inversion tectonics* or *structural inversion* to explain the evolution of the Sole Pit structure in the North Sea, which involved "conversion of a basin area into a structural high". These terms were subsequently used more restrictively by Williams et al. (1989), who

considered that "*structural inversion* (or *inversion tectonics)* occurs when basin-controlling extensional faults reverse their movement during compressional tectonics, and to varying degrees, basins are turned inside out to become positive features". Although inversion is generally assumed to be positive, some authors make a distinction between "positive" and "negative" inversion, with the latter being defined by Williams et al. (1989) as "the reactivation in extension of a significant portion of an existing contractional system". At the same time, Cooper et al. (1989) used the term "*basin inversion*" to describe "a

basin controlled by a fault system that has been subsequently compressed-transpressed producing uplift and partial extrusion of the basin fill", without being specific as to whether pre-existing faults need to reverse their movement or whether new faults are formed. As becomes clear from the above, there is no generally accepted definition of the term "inversion" (see also Buchanan & Buchanan, 1995), although the term is widely used.

In the context of this review of analogue modelling of basin inversion, we define the process of "basin inversion" rather broadly as: *the reversal of subsidence in a (rift) basin due to compressional tectonics, so that the sedimentary infill of the basin is uplifted and/or exhumed, with or without reactivation of previously established normal faults*. Basin inversion has traditionally been considered to cover the inversion of continental basins, prior to necking and continental break-up, which is also the general context of this review paper.

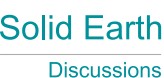
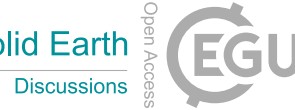

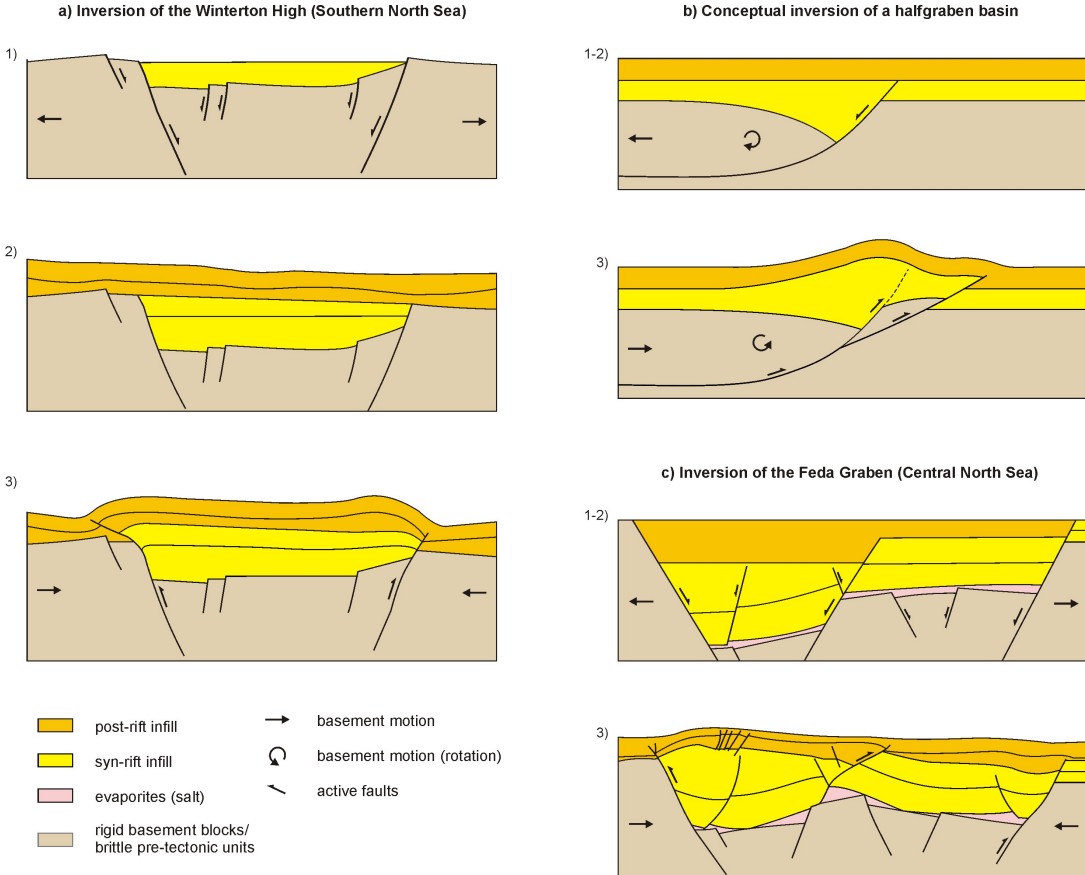

**Figure 1: Examples showing the three main stages of basin inversion: (1) Syn-rift; (2) Post-rift; and (3) Inversion. (a) Formation and inversion of a symmetric graben (Winterton High in the southern North Sea), where bounding normal faults are reactivated**
**and propagate into the overburden at a shallow angle during Cenozoic inversion. Redrawn after Panien et al (2006a), based on Badley et al. (1989), with permission from the Geological Society, London. (b) Schematic example of an inverted half graben related to a listric fault. Inversion causes uplift of the basin fill, and reactivation of the listric fault that propagates upward into the previously unfaulted post-rift units, as well as the development of a new thrust fault (footwall shortcut). Modified after Cooper et al. (1989), with permission from the Geological Society, London. (c) Simplified development of the salt-bearing Feda Graben in the**
**Central North Sea. The salt (Zechstein evaporites) decouples the basin infill from the basement. Based on Gowers et al. (1993) and Stewart & Clark (1999), with permission from the Geological Society, London.**





### 1.2. Importance of basin inversion tectonics

Inverted basins are very common geological features and are found in multiple locations, for instance the North Sea area
(Lamplugh 1991, Glennie & Boegner 1981, Nalpas & Brun 1996; Evans et al. 2003; De Jager 2003, Hansen et al., 2021), the
Atlas Mountains (Vially et al. 1994), the Pyrenees and Alps (Pfiffner 1993; Kiss et al. 2020; Mencos et al. 2015; Lescoutre
& Manatschal 2020; Musso Piantelli et al. in prep), the Araripe basin in NE Brazil (Marques et al. 2014), offshore Korea
(Park et al. 2021) and China (Yu et al. 2021), as well as many other places around the globe (Letouzey 1990; Lowell 1995;
Iaffa et al., 2011; Gibson & Edwards 2020; Bosworth & Tari 2021; Dooley & Hudec 2021).


A thorough understanding of the geological processes involved is not only relevant for science, but is also of great
importance for societal and economic reasons. First, the ongoing tectonic deformation in many inverted basins, which are
often incorporated into active mountain belts, causes seismic hazards that need to be assessed and monitored (Plenefisch &
Bonjer 1997; Edwards et al. 2015; Mock & Herwegh 2017; Madritsch et al. 2018; Deckers et al. 2021). Second, basin
inversion has proven to be a key process for hydrocarbon trap formation in petroleum provinces around the globe and
determining the timing of inversion has been historically crucial to petroleum geologists (De Jager 2003, Evans et al. 2003;
Turner & Williams 2004; De Jager & Geluk 2007; Cooper & Warren 2010 and 2020; Tari et al. 2021, as also demonstrated
in two special volumes edited by Cooper et al. 1989, and Buchanan & Buchanan 1995). Third, since petroleum reservoirs
can subsequently be used for $CO_2$ sequestration (Voormeij & Simandl 2004; Li et al. 2006), knowledge of inversion
tectonics can be applied to mitigate the effects of greenhouse gas emissions. Fourth, basin inversion processes and related
fluid flow may furthermore lead to the development of mineral resources (Sibson & Scott 1998; Liang et al. 2021; Gibson &
Edwards 2020). And last, a thorough understanding of basin evolution, including basin inversion, is also of great interest for
geothermal energy projects (Edwards et al. 2015; Vidal & Genter 2018; Doornenbal et al. 2019; Békési et al. 2020; Weibel
et al. 2020). Future applications, including exploration for natural hydrogen, are presented in section 7.


### 1.3. Analogue modelling of basin inversion

When studying tectonic processes, researchers have a couple of major obstacles to face. Firstly, the size of the systems
involved is massive so that a thorough (structural) mapping is a major challenge. Secondly, large parts of these systems are
simply inaccessible as they are deep below the surface, covered by thick sedimentary sequences or situated far offshore.
Recent advances in mapping techniques and geophysical methods have mitigated these obstacles to a certain degree, yet the
most challenging impediment on the path to a thorough understanding of tectonic processes are the vast timescales involved.
It is simply impossible to directly observe the evolution of a tectonic system in a human lifetime.

To circumvent these limitations, geologists have for over two centuries applied analogue modelling techniques with the aim
of the simulating large-scale tectonic processes at convenient time- and length scales in the laboratory. By using relatively



simple model materials representing the different layers in the lithosphere that are subsequently deformed in experimental apparatus, such large-scale tectonic processes can be reproduced on a small scale, and within a matter of hours or days. In addition to simulating the dynamic aspects of tectonic processes, researchers can also systematically test the influence of
specific parameters on their models and compare the model results to natural examples. As such, analogue modelling is an excellent tool to study the dynamics of tectonic processes and has greatly contributed to our understanding of the evolution of our dynamic planet.

Analogue modelling as a technique to study tectonic processes evolved since the early 19$^{th}$ century (Hall, 1815, Cadell 1888,
see also the reviews by Koyi 1997; Ranelli 2001; Bonini et al., 2012; Graveleau et al. 2012; Zwaan & Schreurs in press). However, the first experimental studies aiming at basin inversion were only performed in the second half of the twentieth century (Lowell 1974; Koopman 1987, McClay & Buchanan, 1992, Mitra 1993, McClay 1992). This relatively late start may have been caused by a late interest in basin inversion processes in general, which only flared up in the 1980's, as well as the relative complexity of these models, which require sophisticated experimental machines capable of simulating multiple
deformation phases. Nevertheless, the field of basin inversion modelling has steadily advanced over the decades, following the same trends as other analogue tectonic modelling fields. These trends include a shift of focus from qualitative to quantitative modelling practices through the use of new and improved model materials, experimental set-ups, monitoring techniques and analyses.

### 1.4. Aim of this review

The main aim of this review is to recap the advances in knowledge of basin inversion tectonics acquired through analogue modelling studies since the previous reviews by McClay (1995) and, more recently, by Bonini et al. (2012). In this work we systematically review over 70 experimental studies of basin inversion, providing an up-to-date summary of the different set-ups, materials, tested parameters, and results. We also assess how these results compare to numerical models of basin inversion tectonics, and to natural examples of inverted basins. We furthermore identify the various perspectives and
opportunities in the field, which we hope will serve as an inspiration for future analogue modelling studies of basin inversion.



### 2.1. Mechanics of basin inversion

For a basin to be inverted, it needs to form a mechanically weak region in comparison to its immediate surroundings.
Mechanical weaknesses may stem from the basin fill, allowing uplift and folding of sedimentary layers upon shortening, and from extensional faults formed during initial basin formation. Basin inversion is generally well discernable in the upper crust, where brittle deformation dominates. The rheology of brittle materials is generally regarded as time-independent, roughly obeying a Mohr-Coulomb criterion of failure (Coulomb, 1773), that describes the relation between the shear stress ($\tau$) parallel to a (potential) fault plane required for fault activation, the stress normal to the fault plane ($\sigma_n$) as well as the
cohesion ($C_0$) and the angle of internal friction ($\varphi$) of the material as follows (Fig. 2a):

$$\tau = C_0 + \sigma_n \, tan \, \varphi \qquad\qquad (1.1)$$

Basin inversion generally implies a change from an extensional setting with the maximum compressive stress, $\sigma_1$, oriented
vertically and the minimum compressive stress, $\sigma_3$, oriented horizontally (Fig. 2a), to a shortening setting with $\sigma_1$ oriented horizontally and $\sigma_3$ vertically (Fig. 2b). As such, Coulomb-style normal faults related to initial basin formation form at Coulomb dip angles of ca. 60º (for a $\varphi = 30º$) (Fig. 2a), whereas reverse faults related to subsequent shortening preferentially form at dip angles of ca. 30º (Fig 2b). It follows that normal faults are, under ordinary circumstances, misoriented for reactivation in shortening.


However, fault reactivation of normal faults in shortening has been observed in nature, and can be explained by four mechanisms. Firstly, normal faults that formed at lower dip angles with the horizontal (e.g., Roscoe or Arthur dip angles, Roscoe 1970, Arthur 1977 or low-angle detachment faults) are more readily reactivated, since they may reach the reactivation envelope earlier for lower stresses than a new thrust fault (Fig. 2c). Secondly, a lowering of fault strength by
fluids or mineral alignment (strain softening, Sibson 1985, 1995 and 2009) can reduce the internal friction angle, thus flattening the reactivation envelope and promoting normal fault reactivation (Fig. 2c). Thirdly, thrusts may form close to normal faults, thus using the normal fault as a heterogeneity, but not strictly reactivating (all parts off) the normal fault (Fig. 1b). Finally, oblique shortening facilitates normal fault inversion. Here the plane containing the maximum and minimum principal stresses $\sigma_1$ and $\sigma_3$ is oriented at an angle to the normal fault, so that the effective fault angle is reduced and part of
the deformation is accounted for by a strike-slip component (Dubois et al. 2002) (Fig. 3).





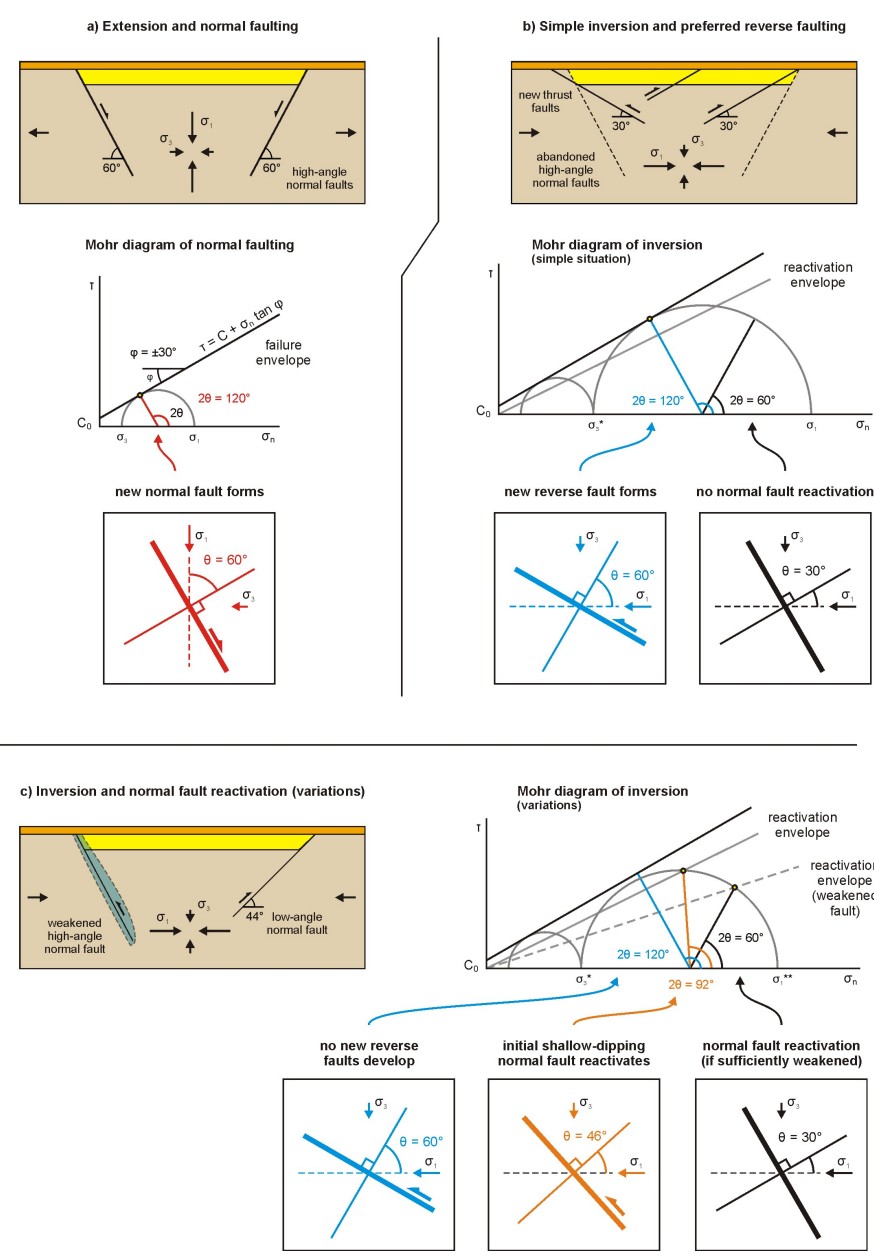



**Figure 2: Schematic 2D illustration of (a) basin development, (b) inversion with preferential new reverse fault development over**
**reactivation of steep normal faults, and (c) inversion variations involving normal fault reactivation due to differences in normal fault dip and fault strength, including schematic Mohr diagrams of the changing states of stress in these systems. $\sigma_1$ = maximum compressive stress, $\sigma_3$ = minimum compressive stress, θ = angle between normal to fault plane and orientation of the maximum compressive stress ($\sigma_1$), φ = angle of internal friction, $C_0$ = cohesion, τ = shear stress, $\sigma_n$ = normal stress. The reactivation envelope represents reactivation of a pre-existing fault plane, and as such is considered to have negligible cohesion ($C_0$ = 0 Pa), and**
**therefore starts from the origin. Note that $\sigma_1$ (vertical loading) during extension becomes $\sigma_3$ during subsequent inversion in 2D (as indicated by the small semi-circle in the Mohr diagram in [b]). Inspired by Bonini et al. (2012).**

It should be stressed that even though the brittle behaviour of the upper crustal layers is generally considered to dominate deformation during inversion, also the ductile parts of the lithosphere can play an important role. Such ductile layers (e.g.,

evaporites, shales, or on a larger scale the lower crust) can decouple different parts of the lithosphere. The degree of decoupling may depend on the viscous layer thickness and distribution, its viscosity as a function of compositional changes throughout a basin, and the tectonic strain rate (Brun 1999). Such decoupling enables the development of significant differences in deformation in the units above and below the viscous layer (thin- vs. thick-skinned deformation). Furthermore, in the case of salt, reactivation of inherited passive diapirs can occur due to major subsalt fault movement. Excellent

examples of the effect of such decoupling can be observed in the North Sea area or the Pyrenees (Stewart & Coward 1995; Stewart & Clark 1999; Stewart 2007; Mencos et al. 2015; Van Winden et al. 2018) (Fig. 1c).

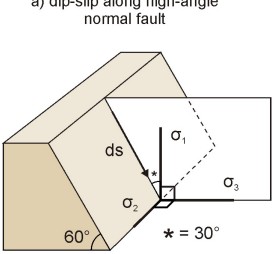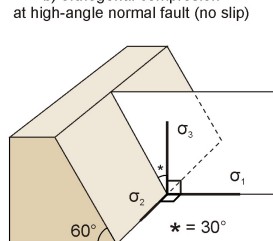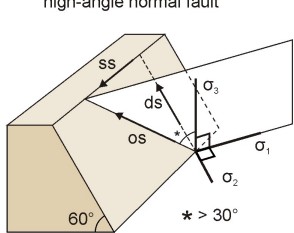

**Figure 3: Oblique reactivation of a high-angle normal fault due to the reduced effective fault angle (✱). Components of deformation: ds: dip-slip, os: oblique slip, ss: strike-slip. $\sigma_1$ = maximum compressive stress, $\sigma_2$ = intermediate compressive stress, $\sigma_3$ = minimum compressive stress. Inspired by Dubois et al. (2002). Note, however, that actual displacements along pre-existing structures can deviate from the directions of the principal stresses (Withjack & Jamison 1986; Morley 2010).**




### 3. Analogue modelling techniques

In this section we address the different model materials, set-ups and scaling principles, as well as monitoring techniques that are used in analogue modelling laboratories around the globe. It may be noted that most of these techniques are very similar to those used for analogue modelling of extensional tectonics (see the reviews papers by Vendeville et al. 1987; McClay 1990 and 1996; Naylor et al. 1994; Koyi 1997; Brun 1999 and 2002; Corti et al. 2003; Bahroudi et al. 2003; Zwaan et al. 2019; Zwaan & Schreurs in press).

### 3.1. Materials and rheology


In analogue modelling studies, brittle and ductile layers in the lithosphere are simulated with various brittle and viscous model materials (Fig. 4). The properties of these model materials are determined in detail using ring-shear testers and rheometers (e.g. Panien et al. 2006b; Klinkmüller et al. 2016; Rudolf et al. 2016; Ritter et al. 2018a,b; Zwaan et al. 2018b).

### 3.1.1. Brittle materials


Granular materials are often used to reproduce the behaviour of brittle parts of the lithosphere (Fig. 4). When deformed, granular materials will form shear zones at similar angles to faults in rocks and soils in nature (see section 3.5). A standard granular material, used to simulate bulk brittle rocks in most basin inversion models included in this review, is quartz sand of
some sort (McClay & Buchanan 1992). Other granular materials include feldspar sand (Munteanu et al. 2013) and corundum sand (Panien et al. 2006b), or mixtures of various granular materials (Abdelmalak et al. 2016; Montanari et al. 2017; Dooley & Hudec 2020). Although these materials may have slightly different properties with respect to quartz sand (notably grain size, density, cohesion and angle of friction), they generally deform in the same fashion. Granular materials should be sieved from a minimum height into the model apparatus to ensure a homogeneous density distribution (Krantz, 1991b; Schellart et
al., 2000; Klinkmüller 2016; Schmid et al. 2020). Importantly, the rheology of granular materials can generally be considered strain rate-independent, even though there are some complexities that can be of importance (Vermeer 1990; Ritter et al. 2016; Montanari et al. 2017; and references therein, section 3.5).

Modellers regularly combine different types of granular materials in their models. Coloured or dyed sand is used to create
(thin) layering that will be visible on side view images or cross-sections (section 3.6). This is generally the same sand as used for bulk brittle rock layers, but in some cases other materials are used, such as corundum sand or Pyrex (all visible on CT-scans, Letouzey et al. 1995; Panien et al. 2006a, section 3.6). These thin horizons of different materials in the bulk sand layers are not considered to significantly affect the bulk model behaviour. By contrast, materials such as micas and microsbeads have a significant lower angle of internal friction and deform more readily. Weaker mica layers are used to



facilitate interlayer slip, while conveniently creating visible layering on sections as well, (McClay 1989, 1996; Buchanan
        1991; Buchanan & McClay 1992, see section 3.6), whereas microbeads have been used to simulate basal detachment layers
        (Panien et al. 2006a). Microbeads can also serve as weak sedimentary infill of a rift basin that may be more easily deformed
        during inversion (Martínez & Cristallini 2017, Panien et al. 2005; Yagupsky et al. 2008).

Other modellers have applied wet clay to simulate brittle rocks (Mitra 1993; Mitra & Islam 1994; Eistenstadt & Withjack
        1995; Eistenstadt & Sims 2005). Clay behaves somewhat differently from granular materials as its behaviour has a strain
        rate-dependent component (Oertel 1965; Eisenstadt & Sims 2005 and references therein), which makes it appropriate for
        modelling the uppermost parts of the lithosphere, creating more intricate fault structures when deformed than sand
        (Eisenstadt & Sims 2005). Similar to their granular counterparts, coloured clay can be used to highlight layering and to
distinguish syn-tectonic infill. Buchanan & McClay (1992) have also used a mixture of clay and sand to simulate layers of
        higher competence. In contrast to sand, clay can readily be modelled into shape, but it's highly important to control its water
        content as this significantly affects the material's rheology (Arch et al. 1988; Eisenstadt & Sims 2005, and references
        therein). This is also true for the wet sand-cement mixtures applied by Mandal & Chattopadhyay (1995).

**3.1.2. Viscous materials**

        In analogue modelling studies, the ductile parts of the lithosphere are generally simulated by means of viscous materials
        (Fig. 4). Typical viscous materials include silicone oils such as Polydimethylsiloxane (PDMS, e.g. SGM 36) and "putties"
        (e.g. Dow Corning 3179 Dilatant Compound, or Rhône-Poulenc Gomme Spéciale GS1R), and various types of mixtures of
such materials with granular materials, acids and other substances (e.g. Schellart & Strak 2016; Zwaan et al. 2020a). When
        deformed, these materials flow rather than forming discrete fault structures, and in contrast to the granular materials, their
        behaviour is strongly strain rate-dependent. Importantly, viscous materials are much weaker than their brittle counterparts,
        and in basin inversion models they are often used to simulate intra-crustal (salt) detachments (Letouzey 1995, Brun &
        Nalpas 1996; Dooley & Hudec 2020), or the weak ductile lower crust and lower lithospheric mantle layers (Cerca et al.
2005; Gartrell et al. 2005; Konstantinovskaya et al. 2007; Mattioni et al. 2007; Munteanu et al. 2013 and 2014) (Fig. 4).
        Viscous model materials used in basin inversion experiments commonly have a near-Newtonian rheology at typical model
        deformation rates, which aims to represent dislocation creep over geological timescales (Weijermars 1986; Weijermars &
        Schmeling 1986; Rudolf et al. 2016, and references therein).




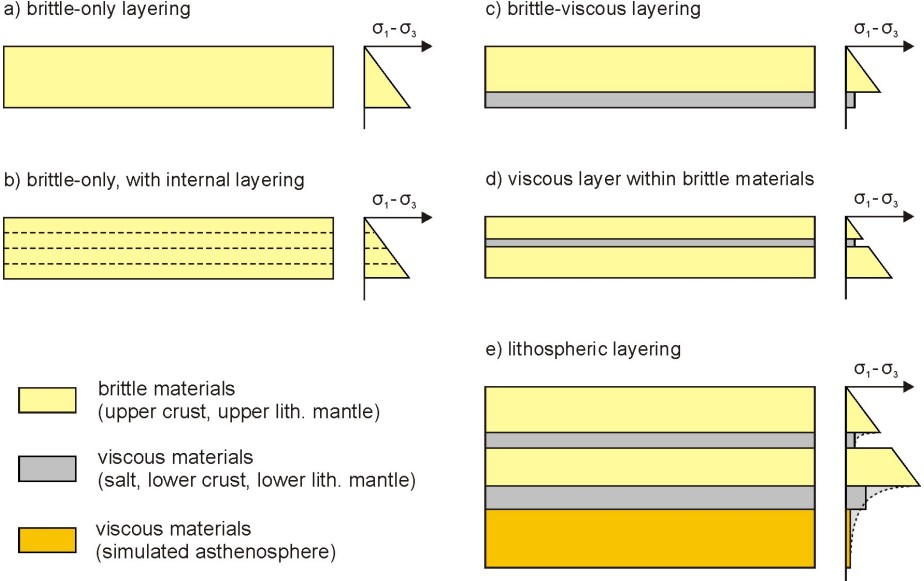

**Figure 4: Examples of model layering used for basin inversion experiments seen in section view, including schematic strength profiles. (a-d) Crustal-scale layering options. (a) Homogeneous layer of brittle model material. (b) Brittle model material**
**interlayered with other brittle materials for visualization or to simulate brittle-style detachments (e.g., Buchanan & McClay 1991). (c) Brittle-viscous layering, with a cover of brittle model material overlying a viscous detachment layer that decouples the brittle cover from the (rigid) model base. (d) Viscous detachment within the brittle model materials (such as a weak shale or salt layer, e.g., Brun & Nalpas 1996). (e) brittle-viscous multilayer arrangement representing the whole of the lithosphere (e.g., Cerca et al. 2005). The dotted line in the strength profile of panel (e) indicates a schematic lithospheric strength profile that is approximated in**
**the model (see e.g., Brun 1999).**

## 3.2. Overview of general basin inversion set-ups

When modelling basin inversion, a basic necessity is choosing a set-up that can induce the type of deformation required for
both the initial extensional phase and the subsequent compressional (inversion) phase (Fig. 5). Both deformation phases are generally induced by rigs that move basement blocks, base plates, rubber sheets or sidewalls/backstops below or into the analogue model materials representing the Earth's lithosphere. An exception are the models by Gartrell et al. (2005) and Konstantinovskaya et al. (2007), who instead applied a system of pulleys and weights to drive deformation (force boundary condition). Most inversion models focus on the crustal-scale, where the set-up may include specific assumptions regarding
the properties of the basement or mantle below (see Zwaan et al. 2019 for a discussion on this topic), but some modellers have also simulated basin inversion on a lithospheric scale (Cerca et al., 2005; Gartrell et al., 2005).





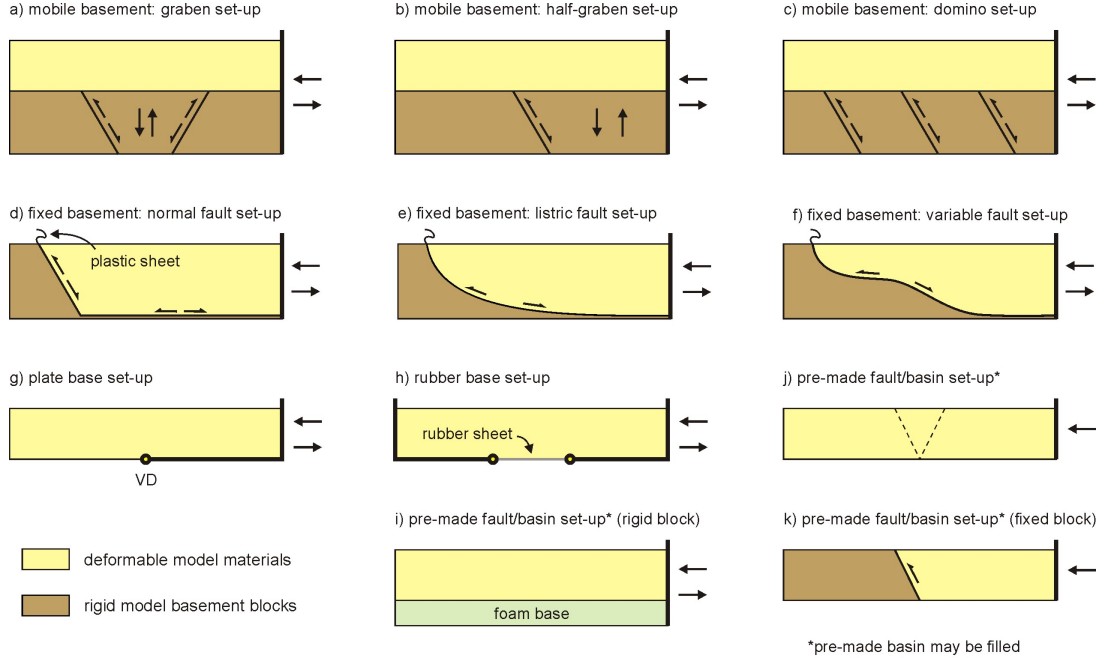

**Figure 5: Examples of basin inversion model setups illustrated in section view. (a-c) Mobile basement block set-ups. (a) Full graben set-up. Note that the graben boundary faults can be set to have different dip angles (e.g., Koopman et al. 1987). (b) Half-graben set-up. (c) Domino block set-up (e.g., Buchanan & McClay 1992). (d-e) Fixed basement (footwall) block set-ups. (d) Steep normal fault set-up (e.g., Buchanan & McClay 1991). (e) Listric fault set-up (e.g., Buchanan & McClay 1991). (f) Variable geometries of the basement block can be applied as well (e.g., McClay et al. 1995; Ferrer et al. 2016). (g) Plate base set-up, with the edge of the plate base inducing a velocity discontinuity (VD) (e.g., Mitra & Islam 1994). (h-i) Distributed deformation set-ups (h) Rubber base set-up, with a rubber sheet spanned between two base plates creating a zone of distributed deformations (e.g., Amilibia et al. 2005). Note that the rubber sheet may also cover the whole base of the model (McClay 1989). (i) Foam base set-up (e.g., Richetti et al. in prep). (j-k) pre-made fault/basin set-up, with either (j) a pre-made fault or basin within the deformable model materials, potentially filled with weaker material (e.g., Panien et al. 2006a), or (k) a basin built next to a fixed footwall block (e.g., Letouzey et al. 1995).**

The first basin inversion experiments by Lowell (1970) and Koopman et al. (1987) involved mobile basement blocks (Fig. 5a). Such set-ups are used to simulate the deformation of a sedimentary cover on top of a rift basin or normal fault developing in the basement (Sanford et al. 1959; Naylor et al. 1994; Dooley et al. 2003). By moving a hanging wall basement block down between two footwall blocks, the overlying model materials are deformed and a basin is generated. This basin can subsequently be inverted by simply moving the hanging wall block upward again (Koopman et al. 1987; Mitra 1993; Mitra & Islam 1994; Burliga et al. 2012; Moragas et al. 2017). Koopman et al. (1987) also applied a set-up to simulate half-graben development above a tilted block (Fig. 5b), and a more complex version of this set-up, used by



Buchanan & McClay (1992), McClay (1995), Jagger & McClay (2018) and Ferrer et al. (in prep), involved a series of

basement blocks that could be tilted simultaneously (domino faulting) to form a broad extensional basin, rather than a single half-graben (Fig. 5c). The basement block motion in both these set-ups can simply be reversed to induce inversion.

Modellers have also regularly used set-ups with fixed basement (footwall) blocks (Fig. 5d-f). In these models, a plastic sheet between the basement block and the model materials is connected to a moving sidewall or backstop, so that the outward

motion of the sidewall caused normal faulting above the footwall block. The resulting rift structures are subsequently inverted by moving the sidewall inward again (McClay 1989, 1995, Buchanan & McClay 1991, Mitra 1993; Mitra & Islam 1994; Gomes et al. 2006, 2010). Alternatively, one can also move the footwall block itself (Yamada & McClay 2004; Ferrer et al. 2016), which is however mostly a change of reference frame (see discussion in Zwaan et al. 2019). Various authors have applied complex footwall block geometries (McClay et al. 1995; Ferrer et al. 2016; Roma et al. 2018a, b) (Fig. 5f), or

different backstop geometries (Gomes et al. 2010).

Base plate or conveyer belt set-ups have been commonly used for modelling extensional tectonics (Allemand et al. 1989; Allemand & Brun 1991; Brun & Tron 1993; Keep & McClay 1997; Michon & Merle 2000, 2003; Gabrielsen et al. 2016; Zwaan et al. 2019) (Fig. 5g). The edge of the base plate or conveyor belt creates a velocity discontinuity (VD), representing

a fault or shear zone in the basement or upper mantle, that localizes deformation in the overlying model materials as the plate is pulled out from under them, resulting in the development of a rift basin (Fig. 5g). Subsequently, the motion of the base plate can be reversed, inducing compression along the VD and in the previously established rift basin (Mitra & Islam 1994; Eisenstadt & Withjack 1995; Nalpas et al. 1995; Brun & Nalpas 1996, Bonini et al. 1998; Dubois et al. 2002; Mattioni et al. 2007; Konstantinovskaya et al. 2007; Pinto et al. 2010). Some models include two plates on both sides of the model that

move apart in opposite directions during both extension and compression (Muneanu et al. 2014). Panien et al. (2005), Del Ventisette et al. (2005, 2006), Yagupsky et al. (2008), Granado et al. (2017) and Miró et al. (in prep.) applied a base plate mechanism to induce a rift basin, which was subsequently inverted by moving a backstop into it.

Other basin inversion set-ups involve distributed basal deformation, for instance by means of a basal rubber sheet. This

rubber sheet is generally inserted between two base plates (Amilibia et al. 2005; Dooley & Hudec 2020; Yu et al. 2021) (Fig. 5h), even though some authors have also used a rubber sheet covering the full length of the model (McClay 1989). By pulling apart the base plates or sidewalls between which the rubber is fixed, the rubber is stretched and creates a distributed type of deformation at the model base, rather than the highly localized deformation induced by a VD (Withjack & Jamison 1986; McClay 1990; McClay et al. 2002; Corti et al. 2007; Henza et al. 2010; 2011; Zwaan et al. 2019). Similar to the base

plate models, the resulting rift structure can be simply inverted by moving the plates (or sidewalls) together again so that the rubber base contracts (Amilibia et al. 2005; Dooley & Hudec 2020). Alternatively, foam or a combination of foam and plexiglass bars can be used to reproduce distributed deformation at the model base (Scheurs & Colletta 1998; Zwaan et al.





2016; Richetti et al., in prep). Instead of being stretched, the foam or foam/plexiglass assemblage needs to be compressed between sidewalls first, and as the model sidewalls are moved apart or together, the foam base extends or contracts in a
distributed fashion, deforming the overlying materials (Zwaan et al. 2019, 2020b; Richetti et al. *in prep*).

Some researchers simplify the modelling procedure by building the basin or normal fault(s) already during model preparation (Fig. 5j), obtaining inversion by subsequently moving a backstop into the model (Sassi et al. 1993; McClay et al. 2000; Panien et al. 2006a; Marques & Nogueira 2008; Di Domenica et al. 2014; Martínez et al. 2016, 2018; Martínez &
Cristallini 2017). Such backstop models can be combined with fixed basement blocks that represent the footwall block of a rift boundary fault or half-graben structure (Vially et al. (1994), Letouzey et al. 1995, Philippe 1995, Fig. 5k). Vially et al. (1994), Letouzey et al. (1995) and Roure & Colletta (1996) have also inverted pre-built basins in a deformable basement block set-up (Fig. 5b), where only inversion of the pre-built basin was applied. Buchanan & McClay (1992) used a domino-rig (Fig. 5c) for inverting a series of pre-built basins instead. It may be noted that these moving sidewall models that include
pre-built basins or faults may overlap to a degree with thrust wedge experiments (Graveleau et al. 2012).

Finally, some modellers have simulated basin inversion on a lithospheric scale, rather than on the more standard (upper) crustal scale (Cerca et al. 2005; Gartrell et al. 2005). Lithospheric-scale modelling of rifting in a normal gravity field (in contrast to centrifuge methods with enhanced gravity conditions, Corti et al. 2003; Agostini et al. 2009; Zwaan et al. 2020a)
is generally done with set-ups involving mobile sidewalls (Allemand et al. 1991; Brun & Beslier 1991; Nestola et al. 2013, 2015; Beniest et al. 2018; Zwaan & Schreurs 2021, in prep). By moving the sidewalls apart, the model, with layers representing the whole lithosphere floating on a dense liquid or weak viscous layer simulating the underlying asthenospheric mantle (Fig. 4e), is stretched. By simply reversing the motion of the sidewalls, rift basins that developed during the initial extension phase may be inverted. The available lithospheric-scale basin inversion model results come from very specific set-
ups (involving complex lithospheric inheritance in the case of Cerca et al. 2005, and the oblique reactivation of a transfer fault system in the case of Gartrell et al. 2005), preventing generalized insights so far. We do therefore not address these models in further detail in this review.

### 3.3. Additional model set-up variations

We described the general model set-ups and model materials in the previous section. However, there are numerous possible variations and adaptations, especially regarding model layering and structural inheritance in both 2D and 3D (Fig. 6).

Even though many basin inversion set-ups are essentially 2D, the three-dimensional nature of tectonic processes is an important consideration for basin inversion models. Plate motion directions change over time, which can lead to changes in
tectonic regimes (Sibuet et al. 2004; Philippon & Corti 2016; Schmid et al. 2017; Brune et al. 2018; Angrand et al. 2020). To





account for such changes in direction between deformation phases, modellers need modelling machines that can reproduce such kinematic changes (Fig. 6a-c). This can be done by simply repositioning the motor that controls the inward and outward motion of the moving parts with respect to the model (Dubois et al. 2002; Nalpas & Brun 1995; Brun & Nalpas 1996; Pinto et al. 2010), or by combining perpendicular and lateral motion to allow for oblique extension and oblique compression

(Schreurs & Colletta 1998; Mattioni et al. 2007) (Fig 6a-c). Some authors have even applied rotational extension and compression in their inversion models (Jara et al. 2015; 2018) (Fig. 6d-f).

A further option to add 3D complexity is the inclusion of different along-strike geometries of base plates, such as oblique VD's (Panien et al. 2005; Ustaszewski et al. 2005; Munteanu et al. 2014; Jara et al. 2015, 2018; Granado et al. 2017, Wang et

al. 2017; Deng et al. 2019), transfer fault structures (Konstantinovskaya et al. 2007, Likerman et al. 2013) and pull-apart systems (Wang et al. 2017, Fig. 6g-i). Similarly, complex 3D variations such as along-strike curving geometries have been applied in basement block set-ups (Yamada & McClay 2003a, b, 2004). Modellers have also tilted the model base, to accomplish complex layered geometries and heterogeneous normal stresses (Philippe 1995; Granado et al. 2017; Borderie et al. 2019).


Additional variations can be made to the general model layering (Fig. 4). Weak granular materials (section 3.1.2) can be used to simulate (basal) detachment layers (Buchanan 1991; Panien et al. 2006a). Viscous materials are often used to simulate weak layers or detachments in basin inversion models (Fig. 4). For instance, Vially et al. (1994), Letouzey (1995), Nalpas et al. (1995), Brun & Nalpas (1996), Dubois et al. (2002), Ferrer et al. (2016) Granado et al. (2017), Roma et al.

(2018a, b) and Dooley & Hudec (2020) added layers of viscous material to their sand pack for simulating weak (salt) intervals in their basin inversion experiments, decoupling the simulated sedimentary cover from the basement units (Fig. 4c, d). Patches of viscous materials are used as a handy method to distribute crustal deformation above an otherwise (too) strongly localizing VD in crustal-scale base plate models (Brun & Nalpas 1996; Del Ventisette 2005, 2006; Sani et al. 2007; Pinto et al. 2010; Likerman et al. 2013; Jara et al. 2015, 2018), even though deformation may also be focussed along the

edges of these patches. Other researchers have applied a viscous layer throughout the model to simulate a ductile lower crustal layer underlying the brittle upper crust (Konstantinovskaya et al. 2007; Mattioni et al. 2007; Bonini et al. 2012; Munteanu et al. 2013, 2014, Fig. 4d).

Furthermore, variations within the model layers allow for the simulation of complex 3D structural inheritance. For instance,

weak granular materials can serve to simulate sedimentary basin infill (Martínez & Cristallini 2017, Panien et al. 2005; Yagupsky et al. 2008). Pre-cut fault planes in brittle layers serve to localize deformation (Panien et al. 2006a, Di Domenica et al. 2014). Marques & Nogueira (2008) even embedded viscous material in such pre-existing faults during model preparation, reproducing the effects of salt injected along a fault plane during deformation. Additional methods to generate



complex weaknesses is the application of patches or "seeds" of viscous material to locally weaken the overlying sand layers,

thus localizing deformation (Munteanu et al. 2013, Dooley & Hudec 2020).

**Oblique inversion tectonics (base plate example)**

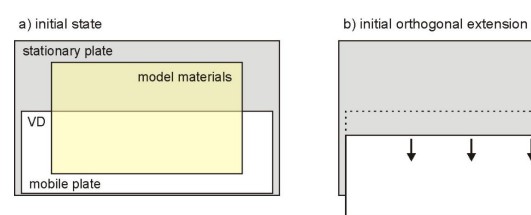

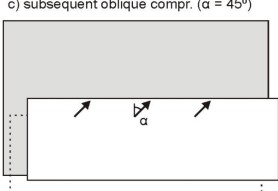

**Rotational inversion tectonics (base plate set-up example)**

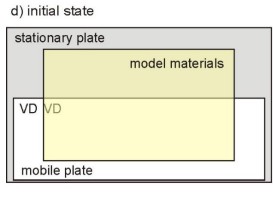

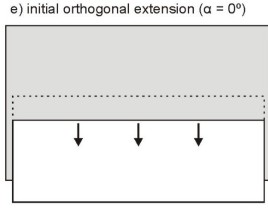

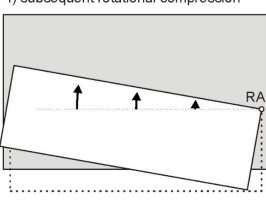

**Pull-apart inversion tectonics (base plate set-up example)**

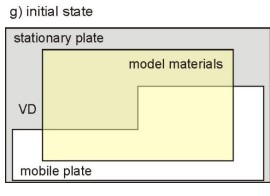

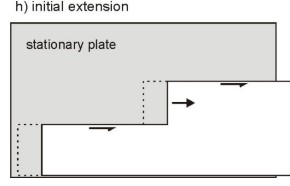

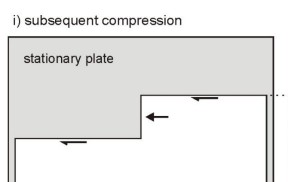

**Figure 6. Examples of 3D kinematics in basin inversion models. (a-c) Oblique inversion tectonics example. The plate motion direction is defined as angle α, the angle between the normal to the model axis and plate motion direction (e.g. Zwaan et al. 2016). VD: velocity discontinuity (mobile base plate edge). Based on Brun & Nalpas (1996). (d-f) Rotational inversion tectonics example. RA: rotation axis. Based on Jara et al. (2015 and 2018). (g-i). Inverted pull-apart basin example. Based on Wang et al. (2017). Modellers have used a wide variety of other base plate geometries for basin inversion experiments (see section 3.3.).**




### 3.4. Inclusion of additional geological processes

Surface processes (i.e., erosion, transport and syn-kinematic sedimentation) can have important impacts on the evolution of both extensional and compressional tectonic systems (Koons, 1990; Burov & Cloetingh 1997; Buiter et al. 2008; Graveleau
et al. 2012; Zwaan et al. 2018a; Borderie et al. 2019), and are thus naturally also of importance for basin inversion. Often, modellers simulate extension by filling in the rift basin with (weaker) brittle model materials during or after the initial extensional deformation phase (Panien et al. 2005; Pinto et al. 2010; Ferrer et al., 2016; Granado et al. 2017; Moragas et al. 2017). In addition, some studies have included the deposition of sediments during the inversion stage (Buchanan & McClay 1991; McClay 1995; Yamada & McClay 2004; Jagger and McClay 2018; Roma et al., 2018a and b). Including
sedimentation in most cases is done by filling in the negative topography formed in the model, either by sieving or by pouring granular materials, although Moragas et al. (2017) included the simulation of prograding sequences. Including erosional processes is however challenging. Either one must determine where erosion is taking place in the model, and how much model material needs to be removed, or one needs to develop a physical method to directly include active erosional and depositional processes in tectonic experiments (including precipitation, erosion and sedimentation, Graveleau et al.
2011, 2015). To our knowledge, these more complex methods have so far not been frequently used for typical basin inversion modelling due to their natural and technical complexity, although Strzerzynski et al. (2021) apply them in a modelling study of inversion of the Algerian Margin.

Magmatism is a further important geological process that has been frequently studied in analogue models (Corti et al. 2003
and 2004; Poppe et al. 2019; Maestrelli et al. 2021). However, so far only few studies have included magmatic processes in basin inversion models (Martínez et al. 2016 and 2018). Magmatism in these inversion models is achieved by injecting vegetable oil from the bottom of the model set-up during the inversion phase.

### 3.5. Scaling

Scaling of analogue models is necessary to guarantee the similitude of a model and its natural equivalent or proptotype
(Hubbert 1937, Ramberg 1981), allowing for accurate model-nature comparisons. Similitude is achieved by ensuring (1) geometrical, (2) kinematic, and (3) dynamic similarity between the model and prototype. Geometrical similarity requires that all distances (length, width, depth, layer thickness) in the analogue model must have the same proportions as the natural example. Kinematic similarity signifies geometric and temporal similarity of the model and prototype, realized through similarity of velocities. Finally, dynamic similarity is established when all forces, stresses and material strengths are properly
translated from the natural example to the model scale. Although it is practically impossible to incorporate all detailed complexities that characterize natural geological settings into a small laboratory experiment, a correct scaling of the dominant factors controlling deformation will allow the scaling criteria to be fulfilled. The geological setting is usually





approximated with a relatively simple model that uses as few parameters as possible to simulate the system in a meaningful way.


### 3.5.1. Scaling of brittle materials

Although in the past analogue modellers have generally assumed that granular materials behave according to the Coulomb failure criterion (Coulomb, 1773) (equation 1.1) with constant frictional properties, these materials show a more complex

behaviour. Shear tests on granular materials reveal an elastic/frictional plastic behaviour with a phase of strain hardening until peak strength, corresponding to shear zone initiation, with a subsequent phase or strain softening until a dynamic-stable strength value is reached; If shearing is paused and subsequently resumed, shear stress increases to a second peak strength (reactivation peak strength), which corresponds to shear zone reactivation and occurs at a lower stress value than the one required for shear zone initiation (Lohrmann et al., 2003; Panien et al., 2006b; Klinkmüller et al. 2016). From these shear

tests, internal friction angles at first peak strength, dynamic-stable strength and reactivation peak strength can be deduced for a particular granular material, with in general highest values of internal friction angle at first peak strength, lowest values at dynamic-stable strength and intermediate values at reactivation peak strength (Panien et al., 2006b; Klinkmüller et al., 2016).

In this context, the brittle behaviour of the upper crust is roughly characterized by angles of internal friction between ca. 30º

and 40º, and cohesion values between 0 and 50 MPa (Byerlee, 1978). In order to be properly scaled, model materials must have similar angles of internal friction as the upper crust as well as an appropriate (low) cohesion value (Abdelmalak et al. 2016). These criteria are met by many granular materials, which have angles of internal friction between ca. 30º and 40º and negligible cohesion (Krantz 1991b; Schellart 2000; Lohrmann et al. 2003; Panien et al. 2006b; Klinkmüller et al. 2016).

Standard granular materials therefore generally produce shear zones that have similar geometries as faults in intact brittle lithosphere (Schellart & Strak 2016, and references therein), ensuring proper geometrical and dynamic similarity between models and nature (Hubbert 1937). Brittle materials used to implement detachments (microbeads) have an angle of internal friction at peak strength of ca. 20º or lower (Panien et al. 2006b; Bonini et al. 2012; Klinkmüller et al. 2016), reflecting the relative weakness of such detachment layers in nature.


Brittle dynamic similarity can furthermore be secured by comparing the dimensionless ratio ($R_s$) between gravitational stresses and cohesive stress of the model and the natural prototype, which, if similar, indicates proper scaling:

$$R_s = \frac{gravitational\ stress}{cohesive\ strength} = \frac{\rho \cdot g \cdot h}{C_o} \qquad (1.2)$$


Here ($\rho$) is density, ($g$) gravitational acceleration, ($h$) a representative length scale and ($C_0$) cohesion.





### 3.5.2. Scaling of viscous materials

In contrast to their brittle counterparts, viscous materials show time-dependent behaviour. When no strain hardening or
softening occurs (as is the case for most viscous materials used in basin inversion models), the material's viscosity remains
constant and its rheology is characterized by Newtonian flow. We can then apply the following formulas to determine the
stress ratio σ* (convention for ratios: σ* = σ $_{model}$/σ $_{nature}$): (Weijermars & Schmeling 1986):

$$\sigma^* = \eta^* \cdot \acute{\epsilon}^* = \rho^* \cdot g^* \cdot h^* \qquad\qquad\qquad \text{(1.3 and 1.4)}$$


Here $\acute{\epsilon}^*$ is the strain rate ratio, and h* the viscosity ratio. Subsequently, the velocity ratio (v*) and the time ratio (t*) are
obtained so that a deformation rate or a timespan in the laboratory can be translated to their respective values in nature and
vice versa:

$$\acute{\epsilon}^* = \frac{v^*}{h^*} = \frac{1}{t^*} \qquad\qquad\qquad\qquad \text{(1.5 and 1.6)}$$

In order to secure proper dynamic similarity, the dimensionless Ramberg number ($R_m$), involving the ratio between
gravitational stress and viscous stress of the model and its natural equivalent can be compared (Weijermars & Schmeling
1986):


$$R_m = \frac{gravitational\ stress}{viscous\ stress} = \frac{\rho \cdot g \cdot h}{\acute{\epsilon} \cdot v} = \frac{\rho \cdot g \cdot h^2}{\eta \cdot v} \qquad\qquad \text{(1.5)}$$

### 3.5.2. Typical scaling parameters for inversion models

Although every basin inversion modelling study has its own specific scaling parameters, these parameters generally fall in a
clear range (which are in fact quite typical of analogue models in general, and are summarized in Table 1). Basin inversion
models are generally several decimetres, up to perhaps a meter in size (width and length), with model layer thicknesses in the
order of several to perhaps tens of centimetres. A cm in these models may represent one to several km in nature, and model
material densities are often in the order of 1000-2000 kg/m$^3$, whereas rock densities range between 2300-3000 kg/m$^3$. Basin
inversion models mostly involve viscous materials with viscosities in the order of $10^3$ to $10^5$ Pa s, whereas weak ductile
layers in the upper crust may have viscosities in the order of $10^{14}$ Pa s to $10^{18}$ Pa s (e.g. evaporites and shales, e.g. Warren
2016), and ductile parts of the lower crust have viscosities between $10^{19}$ Pa s and $10^{23}$ Pa s (e.g. Buck 1991; Warren 2016).
Deformation rates in terms of imposed sidewall or base plate displacements are generally a few mm to a couple of cm per
hour, which for models involving viscous layers scales to some mm to over a cm per year in nature, well in line with tectonic
displacements observed in nature (e.g. ArRajehi et al. 2010; Saria et al. 2014). Note that the deformation rate can be varied at
will for brittle-only models due to the strain-rate independent behaviour of brittle model materials.





**Table 1. Typical scaling parameters in analogue models of basin inversion**

| | Parameters | | | |
|---|---|---|---|---|
| | Quantity | Unit | Model | Nature |
| Material properties (brittle) | Model density (ρ)* | kg/m$^3$ | 1000-2000 kg/m$^3$ | 2000-3000 kg/m$^3$ |
| | Grain size range (ø) | m | 50-300 μm | - |
| | Internal friction angle (φ) # | | 31˚-40˚ | 30˚-40˚ |
| | Cohesion (C$_{0)}$) | Pa | < 100 Pa | ca. 10$^7$ Pa |
| Material properties (viscous) | Density (ρ)* | kg/m$^3$ | 1000-2000 kg/m$^3$ | 2000-3000 kg/m$^3$ |
| | Viscosity (η) $ | Pa s | 10$^3$-10$^6$ Pa s | 10$^{14}$ Pa s (salt) 10$^{23}$ Pa s (lower crust) |
| | Rheology § | | Newtonian (n ≈ 1) | Newtonian (n ≈ 1) |
| Model geometry and kinematics | Length (l) | m | 10-100 cm | 10-100 km |
| | Deformation rate (v) ¥ | m/s | 0.5-20 cm/h | 0.5-5 mm/yr |
| | Gravitational acceleration | m/s | 9.81 m/s | 9.81 m/s |


\*        Bulk material density can vary between 1000 and 4000 kg/m$^3$. The porosity of granular materials makes a big difference, as does the water content in clays, and the model preparation method (e.g. sieved, poured, scraped).

\#        This includes internal friction angles at peak strength, dynamic-stable strength and reactivation strength for most granular materials, excluding very well-rounded granular materials such as microbeads, which have much lower values. Note that the often used
friction coefficient (μ) is defined as: tan (φ)

\$        May depend on strain rate, if the material deviates from (near-) Newtonian rheologies

§        Generally used analogue materials show (near-)Newtonian behaviour (silicone or PDMS), where n ≈ 1. In nature, this represents the dislocation creep deformation mechanism, valid for gradual deformation over geological time periods (Rudolf et al. 2016 and references therein).

¥        most relevant for the scaling of viscous materials, as the rheology of granular materials is considered to be generally strain rate-independent



### 3.6. Model monitoring and analysis


A key part, and the great strength of any analogue modelling study, is the monitoring and quantification of model deformation over time. Since the dawn of analogue tectonic modelling, researchers have developed a variety of techniques, ranging from (time-lapse) photography, the generation of cross-sections and topography analysis, to advanced 2D and 3D digital image correlation techniques, and X-ray CT-scanning. Most studies included a combination of these techniques.]


### 3.6.1. Photography

Photography is a trusted method for analogue model monitoring, and especially time-lapse photography provides an excellent first-order insight into model deformation. Model monitoring through photography in basin inversion studies can be split into top view and side view approaches (Fig. 7a, b). Many modellers routinely apply top view photography, where

lighting is set to cast shadows that highlight surface structures, and where a surface grid serves to trace deformation (Brun & Nalpas 1996; Del Ventisette et al. 2005; Sani et al. 2007; Panien et al. 2005; Yagupsky et al. 2008; Wang et al. 2017) (Fig. 7a). Such top view time lapse imagery of the model surface allows for statistical fault orientation analysis (Jara et al. 2015) and fault length/displacement analysis (Keller & McClay 1995).

In addition to top view photography, model set-ups with transparent sidewalls enables direct monitoring of model deformation at the sides (Koopman 1987; McClay 1989 and 1995, Buchanan & McClay 1992; McClay & Buchanan 1992; Gomes et al. 2006 and 2019, Jagger & McClay 2016, Fig. 7b). In these cases, sidewall friction may cause boundary effects, but this can be mitigated by using products like Rain-X spray (for car windshield treatment, Krantz 1991a; Herbert et al. 2015), or transparent Teflon foil (Cruz et al. 2010). In some models involving clay, no transparent sidewalls were needed as

the clay was stable enough to not deform under its own weight (Mitra 1993; Mitra & Islam 1994). By adding layers and other markers in section view, a first-order quantification of deformation becomes possible (Marques & Nogueira, 2008; Mitra 1993, Mitra & Islam 1994, McClay et al. 1995, Fig. 7b).

### 3.6.2. Model sectioning

Making cross-sections is another straightforward and popular method to analyse the final stages of internal model deformation (Eisenstadt & Withjack 1995; Brun & Nalpas 1996; Dubois et al. 2002; Amilibia et al. 2005; Del Ventisette et al. 2005, 2006; Konstantinovskaya et al. 2007; Munteanu et al. 2014; Dooley & Hudec 2020, Fig. 7c). Sectioning of sand is normally done by wetting the sand to stabilize it. If required, the wet sand can be frozen for extra stability (Cerca et al. 2005), which has the additional advantage that any viscous materials will be stiffer and thus more stable too. Alternatives are

to impregnate the sand with additives (McClay & Buchanan 1992), for instance with hot gelatine (Jara et al. 2015). Cutting



sections can be done manually using knives, carboard cutters or saws, or automatically with a slicing machine (Ferrer et al., 2016; Jagger & McClay 2018; Dooley & Hudec, 2020). Differently coloured layers allow for an assessment of the model's internal deformation (Fig. 7c).

A drawback of making cross-sections is that the model will have to be destroyed, so that sectioning can only be done at the end of the model run. A clever workaround is presented by Burliga et al. (2012), who sectioned only part of the model to obtain a continuous evolution, but this will only work in a model with no lateral variations in its set-up. Yamada & McClay (2003a, b and 2004) simply ran the same model set-up multiple times, cutting different sections in different orientations in these different models, including horizontal sections (which was also done by Deng et al. 2019). Such horizontal sections

can be used to create isopach maps (Yamada & McClay 2004). Further advanced analysis through sectioning is presented by McClay (1996), Ferrer et al. (2016), Granado et al. (2017), Roma et al. (2018a, b), Dooley & Hudec (2020) and Ferrer et al., (in prep), who used fine-spaced cross sections of constant thickness made with slicing machines to construct 3D voxel images and pseudo-seismic volumes. This method allows a unique interpretation of 3D internal model structures, in a very similar fashion to the analysis of 3D seismic surveys (Fig. 7d).


### 3.6.3. Topography analysis

Where top view imagery provides a first impression of surface deformation, detailed topography analysis allows quantified insights into surface deformation (Fig. 7e). Various options are available, such as 3D digital image correlation (DIC) image

analysis (Dooley & Hudec 2020; Schmid et al., 2021, section 3.6.4), photogrammetry on stereoscopic photographs (Maestrelli et al. 2020, 2021; Zwaan et al. 2020a, 2021a, b), and fringe projection analysis (Barrientos et al. 2008; Martínez et al. 2016). The technique that has been generally used for basin inversion modelling is surface scanning using laser or white light scanners (Bonini et al. 2012; Likerman et al. 2013, Jara et al. 2015, 2018; Granado et al. 2017; Deng et al. 2019, Fig. 7e). Scanning generates digital elevation models that can be processed in GIS software, allowing for instance the

extraction of topographic profiles over time (Jara et al. 2015; Reitano et al. 2020, 2022).


a) Top view analysis

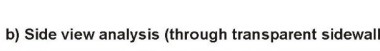

b) Side view analysis (through transparent sidewall)

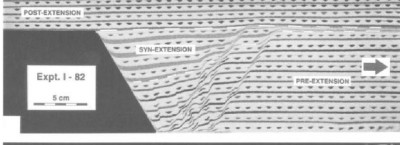

c) Model sections

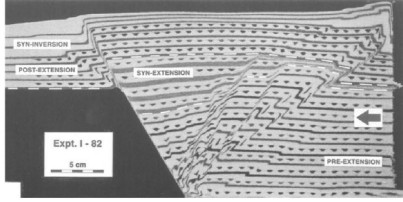

d) Topography analysis

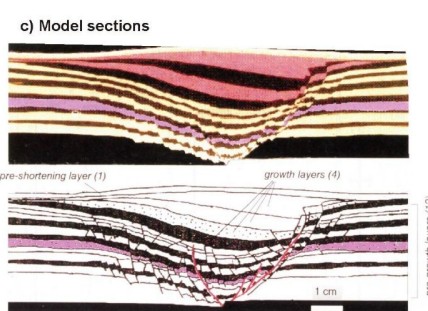

e) 3D section analysis

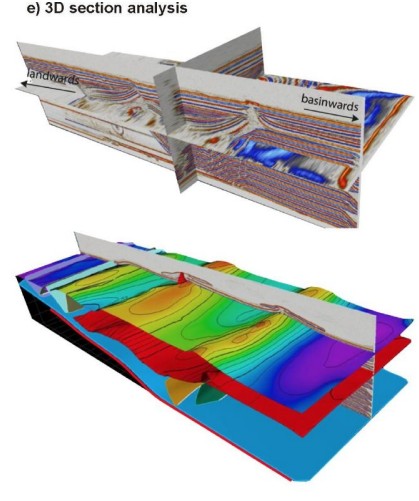

f) DIC analysis

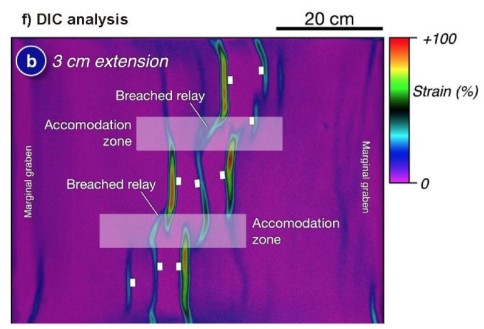

f) X-ray CT analysis

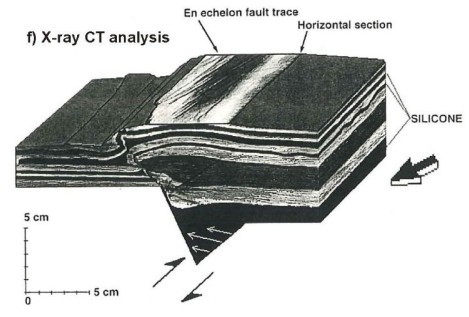





**Figure 7. Examples of techniques used for the analysis of basin inversion models. (a) Top view image from an inversion model by
Panien et al. (2005). Reproduced with permission from Elsevier. (b) Side view imagery of an inversion model with a fixed rigid
footwall block set-up, after McClay (1995). Reproduced with permission from the Geological Society, London. (c) Cross section of
an inversion model with a base plate set-up, after Eisenstadt & Withjack (1995). Reproduced with permission from the Geological
Society, London. (d) Topography map of a basin inversion model by Deng et al. (2019), obtained via surface laser scanning. (e)
Advanced analysis of model cross-sections in 3D seismic interpretation software by Roma et al. (2018b). Reproduced with
permission from the Geological Society, London. (f) Strain map of a basin inversion model by Dooley & Hudec (2020), obtained
through digital image correlation (DIC) analysis. (g) X-ray CT-analysis of an inversion model involving a mobile basement block.
After Letouzey et al. (1995). AAPG ©1995, reprinted by permission of the AAPG, whose permission is required for further use.**

### 3.6.4. Digital image correlation (DIC)

Time lapse imagery of analogue models allows for the quantification of model surface deformation through digital image
correlation (DIC) techniques (Adam et al. 2005; Boutelier et al. 2019). This method compares images from different
moments in time, to derive surface displacement in map view, or when multiple view angles are available, in three
dimensions. The surface displacement field can subsequently be used to extract the amount of strain, and even the type of
faulting (Broerse et al. 2021; Krstekanić et al. 2021). Furthermore, the 3D displacement field provides an alternative method
for topography analysis (Schmid et al., 2021). Even though DIC analysis has recently become a standard in the analogue
modelling toolbox, it has only been sparsely used for basin inversion models. So far only Wang et al. (2017) and Richetti et
al. (in prep) have applied 2D DIC on surface imagery of their inverted pull-apart basin experiments and in their oblique
inversion experiments, respectively, whereas Dooley & Hudec (2020) used 3D DIC analysis (Fig. 7f). Furthermore, Jagger
& McClay (2018) have gone beyond the tracing of markers on sideview imagery (section 3.6.1) and have applied DIC
analysis for analysing deformation on such imagery.

### 3.6.5. X-Ray CT scanning

Most model monitoring techniques (top view imagery DIC and surface laser scanning) can only provide insights into surface
model deformation. Cross-sections allow us to catch a glimpse of internal model structures, but only at the final stage of
model deformation, as the model will have to be physically cut. Although side-view imagery through a transparent sidewall
allows direct observation of model deformation, these observations are only in 2D section view. So far, the only practical
method to obtain 3D insights into internal model evolution is through X-ray computed tomography (CT) methods (Richard
1989; Colletta et al. 1991; Schreurs et al. 2003, Zwaan et al. 2018a, Schmid et al. 2021, Fig. 7g). Such CT-scanning, which
uses X-rays to image the internal structures of a model, has been used in a number of basin inversion studies (Sassi et al.
1993; Vially et al. 1994; Letouzey et al. 1995; Roure & Colletta 1996; Panien et al. 2005, 2006a; Mattioni et al. 2007). Not
only does CT imagery allow unique visualization of 3D model-internal structures, it can also be used to extract specific
horizons in the models (Konstantinovskaya et al. 2007). These horizons can, similar to normal cross-sections, be highlighted
by using layers of different densities that appear in different grey shades on the CT scans (Letouzey et al. 1995).



Furthermore, in a similar way to the side view imagery obtained through a transparent sidewall (section 3.6.1), we can trace material pathways by following markers included in the models (e.g. iodine powder in Panien et al. 2005).

**4. Overview of representative modelling results**

In this section we present overviews of modelling results that are representative of the general structures obtained in analogue models of inversion tectonics (Figs. 8-16). The overviews are categorized in sub-sections that address the results obtained by each of the major types of model set-ups presented in section 3.2 (mobile or fixed basement blocks, base plate, distributed basal deformation, and pre-built basin/fault), with attention to the general influence of model layering, and 3D

factors such as oblique inversion. It is also important to emphasize that analogue modelling results can vary significantly due to variations in material properties and handling techniques (e.g. Schreurs et al. 2006 and 2016, section 3), different degrees of sedimentation, or different amounts and rates of extension and subsequent compression. The vast range of possible variations cannot always be fully accounted for in our generalized overviews. Furthermore, not all combinations of parameters have been tested so far, leading to gaps in the overviews. Hence, we urge the reader to use these overviews as a

first-order guide only, and we refer to the original research for more details.

**4.1. Mobile basement set-ups**

Mobile basement set-ups were among the first set-ups used for basin inversion (Lowell 1974; Koopman et al. 1987; Mitra & Islam 1994), and are normally used for orthogonal inversion experiments (Figs. 5a-c, 8). In the case of a full graben set-up

with a brittle cover (Fig. 8a), initial extension and downward motion of the central rift wedge block leads to the development of a symmetrical graben structure and the creation of accommodation space that can be filled with syn-rift sediments (Fig. 8b). When applying subsequent shortening, the central basement block moves upward again, and the rift boundary faults reactivate (Fig. 8c). However, new low-angle thrust faults, also known as *footwall shortcuts*, develop in the brittle cover, so that only part of the inversion is accommodated by reactivation of the original rift boundary faults. A similar result is

observed in half-graben models with a brittle cover (Burliga et al. 2012, Fig. 8d-f). However, Letouzey et al. (1995) noted that applying oblique inversion in such models promotes the reactivation of the normal faults inherited from the extension phase.



**Overview of inversion models with mobile basement set-ups**

| Initial set-up (section view) | Extension phase | Inversion phase |
|---|---|---|

**Full graben, brittle-only**

a)     b)     c)

**Half-graben, brittle-only**

d)     e)     f)

**Half-graben, brittle-viscous I**

g)     h)     i)

**Half-graben, brittle-viscous II**

j)     k)     l)

**Domino set-up, brittle-only**

m)     n)     o)

| | | |
|---|---|---|
| ▨ rigid basement blocks | ▨ syn-rift infill | → basement motion (translation) |
| ▨ brittle pre-tectonic units | ▨ post-rift infill | ↻ basement motion (rotation) |
| ▨ viscous pre-tectonic units | | ＼ active faults |






**Figure 8. Section-views sketches of idealized results from basin inversion models involving a set-up with mobile basement blocks. (a-c) Full graben set-up with brittle-only layering. (d-f) Half-graben set-up with brittle-only layering, with very similar inversion structures as those in the full graben set-up. (g-i) Half-graben set-up I with brittle-viscous layering (high brittle-to-viscous thickness ratio). (j-l) Half-graben set-up II with brittle-viscous layering (low brittle-to-viscous thickness ratio). (m-o) Domino set-**

**up with brittle-only layering. Images inspired by Koopman et al. (1987), Buchanan & McClay (1992) and Burliga et al. (2012).**

A viscous layer overlying the basement blocks detaches the brittle overburden from the mobile basement and distributes the deformation induced by the basement fault (Fig. 8g-i). As a result, initial extension leads to the formation of multiple faults away from the basement fault (Fig. 8h), and subsequent inversion may only reactivate one of these faults (Fig. 8i) (Burliga et

al. 2012; Moragas et al. 2017). However, the (relative) thickness of the viscous layer and brittle cover has a strong influence on the coupling between basement and cover during extension (Withjack and Callaway, 2000). The structural evolution of such brittle-viscous inversion models can thus vary significantly. As also shown in previous modelling studies (Withjack and Callaway, 2000; Dooley et al. 2003; Moragas et al., 2017; Zwaan et al. 2020a) a relatively thin brittle cover may be subject to flexure during extension (Fig. 8j-l) (for more details see Burliga et al. 2012). The impact of layer thickness on brittle-only

systems should be much less important, as coupling is always high in such a system.

Finally, in models with domino set-ups as used by Buchanan & McClay (1992), McClay (1995) and Jagger & McClay (2018) a series of basement blocks is rotated (Fig. 8m-o), leading to the development of a series of half-grabens with growth strata during extension (Fig. 8n). Inversion of these half-grabens causes partial reactivation of initial normal faults, but also

results in the development of footwall shortcuts (Fig. 8o). Similar to the (half-graben) models described above, these domino set-ups involve inversion along existing normal faults and newly formed thrusts (Fig. 8c, f, o).

### 4.2. Fixed basement set-ups

Similar to the mobile basement set-ups, fixed rigid basement (footwall) block set-ups have been used for some of the earliest

basin inversion models (McClay 1989, 1995, Mitra 1993; Mitra & Islam 1994; Buchanan & McClay 1991) These primarily serve to study inversion as a 2D process with orthogonal extension followed by orthogonal compression (Figs. 5d-f, 9). The edge of the rigid basement block, representing a pre-existing basin boundary fault, can have various geometries (Figs. 5d-f, 9). It may be noted that the applied method of inversion (either by moving the backstop or moving the basement block relative to the model) can cause some variation in structural evolution. The description of these variations is beyond the

scope of this review and we refer the reader to the original publications for more details.

The most straightforward example is the steep normal fault set-up, with brittle-only materials (Buchanan & McClay 1991; McClay 1995; Ferrer et al. 2016, Fig. 9a). As extension is applied by either moving the sidewall and the plastic sheet



between the model materials and the rigid block away, or by moving the rigid block itself away underneath the model
materials, normal faulting is induced (Fig. 9b). In this type of model, a series of normal faults dipping towards the basement
block develops, as well as a major normal fault along the basement block that accommodates most subsidence (Fig. 9b).
Some minor tilting of layers towards the basement block may occur. Inversion of this system slightly reactivates the normal
faults from the initial extensional phase, but inversion is mostly accommodated by the major fault along the basement block,
as well a major backthrust, effectively creating a pop-up structure (Fig. 9). Buchanan & McClay (1991) also show how, in
the presence of a thick post-rift/syn-inversion sequence, the major fault along the basement block can change to a shallower
angle when propagating into these shallower sequences (footwall cut-off).

As seen in the mobile basement models, adding a viscous layer into the pre-rift sequence can detach different parts of the
model. Although no models with a basal viscous layer are known from the literature, some researchers have included a
viscous layer within the brittle materials of a fixed rigid basement model with a steep normal fault (Ferrer et al. 2016, Roma
et al. 2018a, b, Fig. 9d-f). During rifting, the uppermost brittle units are detached from the lower faulted units and form a
salt-detached ramp-syncline basin that is filled with syn-rift units (Fig. 9e). The continuity of the viscous layer or its welded
equivalent inherited from the extensional episode will be critical during the inversion. Inverting this system forms a pop-up
structure in the units below the viscous layer and inverts the ramp-syncline basin (Fig. 9f). The viscous layer acts as an
efficient detachment during inversion and part of the contractional deformation can be propagated above the footwall of the
major fault (Ferrer et al., 2016). Importantly, the flow of the viscous layer causes welding near the edge of the basement
block, and to swell in other areas (Fig. 9f). The final structure is quite distinct from its counterpart without a viscous layer
(compare Fig. 9c with Fig. 9f).

Another well-studied fixed rigid basement block set-up involves a listric fault geometry (McClay 1989, 1995, 1996;
Buchanan & McClay 1991; McClay & Buchanan 1992; Keller & McClay 1995; Gomes et al. 2006, 2010, Ferrer et al., 2016,
Fig. 9g-i). This set-up, generally leads to the formation of a graben at some distance from the top edge of the basement
block, and results in strong tilting of layers near the block (Fig. 9h). During inversion, this tilting is reversed, and similar to
its steep normal fault equivalent, a backtrust develops so that the overall final structure can again be described as a pop-up
(Fig. 9i). Some of the other normal faults may be slightly reactivated as well. (Fig. 9i). The listric fault results described here
are typical of models involving a vertical backstop. Gomes et al (2010) showed that different backstop geometries can also
strongly affect reactivation (e.g., by promoting backthrusting). Furthermore, thin-skinned deformation, simulated by only
having the upper part of the vertical backstop move inward (from either direction in section view) significantly alters the
model results (Gomes et al. 2006), as is also known from thrust wedge experiments (Graveleau et al. 2012).




## Overview of inversion models with fixed basement set-ups

| Initial set-up (section view) | Extension phase | Inversion phase |
|---|---|---|

**Steep fault, brittle-only**

a)      b)      c)

**Steep fault, brittle-viscous**

d)      e)      f)

**Ramp-flat-ramp, brittle-only**

g)      h)      i)

**Ramp-flat-ramp, brittle-only**

j)      k)      l)

**Ramp-flat-ramp, brittle-viscous**

m)      n)      o)

Legend:
- rigid basement blocks
- brittle pre-tectonic units
- viscous pre-tectonic units
- syn-rift infill
- post-rift infill
- basement motion (translation)
- rotation of model materials
- active faults





**Figure 9. Section-views sketches of idealized results from basin inversion models involving a set-up with fixed basement blocks. (a-c) Inversion model with steep fault set-up and brittle-only model materials. (d-f) Inversion with steep fault set-up and a brittle-viscous layering. (g-i) Inversion model with a listric fault set-up and brittle-only model materials. (j-l) Inversion model with a ramp-flat-ramp fault set-up and brittle-only model materials. (m-o) Inversion with a variable fault dip set-up and a brittle-viscous layering. Image inspired by McClay (1989, 1995), Buchanan & McClay (1991), Gomes et al. (2006, 2010), Ferrer et al. (2016).**

Various authors (McClay 1989, 1995; Buchanan & McClay 1992; Ferrer et al. 2016; Roma et al. 2018a, b) have tested the effects of more complex fault geometries, of which a version with a ramp-flat-ramp geometry has been most popular (Fig. 9j-l). Although the main fault has a similar smooth shape to the main fault in the listric fault set-up, leading to similar tilting of layers near the basement block, the flat part in the middle causes a disturbance in the deformation field that can cause local reverse faulting during extension (Fig. 9k). When inverted, the tilted strata near the main fault are back-rotated, but typical of these (brittle-only) models is the development of a pop-up structure at the tip of the flat part (Fig. 9l). Similar to its steep fault equivalent (Fig. 9d-f), adding a viscous layer into the brittle layers of the variable fault set-up decouples the brittle materials below it from those above it, resulting in a sag-like syn-rift deposition pattern during extension (Ferrer et al. 2016; Fig. 9n). When considering the model parts below the viscous layer, inversion creates similar structures as in the brittle-only models, but the flow of the viscous layer creates a smooth inverted basin above it (Fig. 9l, o). Similar to the steep fault model with a viscous layer, the viscous layer swells and is welded here too (Fig. 9f, o). For more insights into the complex interplay between basement fault geometry and brittle-viscous layering see Ferrer et al. (2016) and Roma et al. (2018a, b).

Finally, even though most fixed basement set-up models were designed to investigate 2D inversion, Yamada & McClay (2003a, b, 2004) also explored the third dimension. In these 3D experiments, they applied sinusoidal along-strike variations of the fault surfaces, which induced complex structures in both the initial extension and the subsequent inversion phase. An example of their model results is shown in Fig. 10.



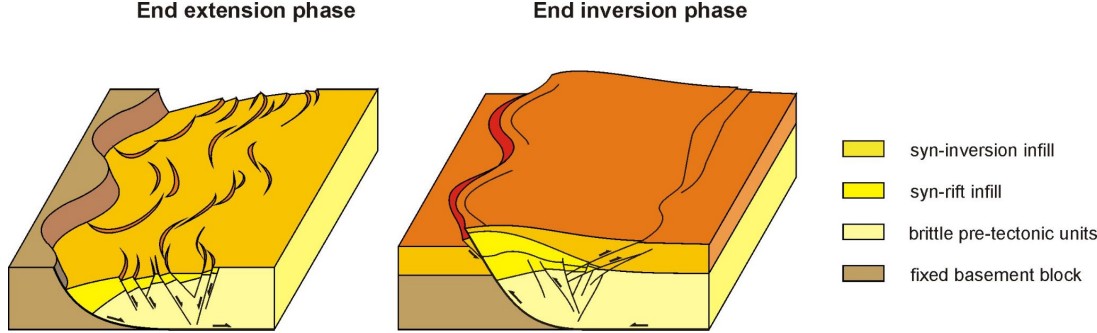


**Figure 10. 3D drawing of inversion model results from Yamada & McClay (2004). AAPG ©2004, image modified by permission of the AAPG whose permission is required for further use.**

**4.3. Base plate set-ups**

Base plate set-ups allow the localization of deformation along the edge of the base plate (velocity discontinuity, or VD) and have been regularly used for orthogonal (2D) as well as 3D inversion (Mitra & Islam 1994; Eistenstadt & Withack 1995; Nalpas et al. 1995, Brun & Nalpas 1996; Bonini et 1998; Eisenstadt & Sims 2005) (Fig. 11). Some authors applied a base plate mechanism to induce a rift basin, which was subsequently inverted by moving a backstop into it (Panien et al. 2005, Del Ventisette et al. 2005, 2006, Sani et al. 2007; Yagupsky et al. 2008, Bonini et al. 2012, Likerman et al. 2013, Munteanu

et al. 2014, Jara et al. 2015, 2018, Martínez et al. 2016, 2018 and Granado et al. 2017). These models can be considered as deforming a pre-built basin by a backstop only and are therefore described in section 4.5.

        In base plate inversion models involving only brittle materials, the use of a VD leads to the development of a graben above the VD, which may become asymmetrical as extension progresses (Allemand & Brun 1991; Ferrer et al. 2016) (Fig. 11b).

Inverting this system in 2D causes reverse faulting starting from the VD, and the formation of a pop-up structure, with only very limited reactivation of previously formed normal faults (Eisenstadt & Withjack 1995; Eisenstadt & Sims 2005) (Fig. 11c). Even so, some modelling studies (Nalpas et al. 1995; Brun & Nalpas 1996) have shown that applying a high enough degree of oblique compression during inversion will (preferentially) reactivate the pre-existing normal faults in this type of experiment (Fig. 11m-o). However, Mitra & Islam (1994), who applied clay instead of granular materials, showed that (2D)

inversion could also be accommodated by large-scale folding, without reactivation of pre-existing faults or nucleation of new thrust faults. The differences between granular materials and clay (in this type of models) are excellently illustrated by the comparative model study by Eistenstadt & Sims (2005).

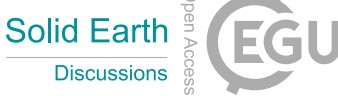

## Overview of inversion models with base plate set-ups

| Initial set-up (section view) | Extension phase | Inversion phase |
|---|---|---|

**Brittle-only**

a) VD  b) →  c) ←

**Brittle-viscous I**

d)  e) →  f) ←

**Brittle-viscous II**

g)  h) →  i) ←

## Orthogonal vs. oblique inversion (3D view)

| After extension | Orthogonal inversion | Oblique inversion |
|---|---|---|

**Orthogonal extension, brittle-only**

j)  k) → α = 0°  l) ← α = 45°

**Orth. extension, brittle-viscous I**

m)  n) → α = 0°  o) ← α = 45°

Legend:
- brittle pre-tectonic units
- viscous pre-tectonic units
- syn-rift infill
- post-rift infill
- → basement motion (translation)
- ↘ active faults



**Figure 11. Sketches of representative results from basin inversion models involving a set-up with base plates. VD: velocity discontinuity. (a-c) Brittle-only inversion experiment. (d-f) Brittle-viscous set-up I with a viscous layer within the brittle layer. (g-i) Brittle-viscous set-up II with a viscous base layer. (j-o) 3D effect of the inversion direction on reactivation of pre-existing faults. Oblique compression promotes oblique-slip reactivation of steep normal faults that are unlikely to be reactivated in an orthogonal compression situation. This effect is also observed in inversion models with a viscous basal detachment layer (Pinto et al. 2010). Images inspired by Dubois et al. (2002), Nalpas & Brun (1993), Nalpas et al. (1995), Brun & Nalpas (1996), Mattioni et al. (2007).**

Nalpas et al. (1995) and Brun & Nalpas (1996) tested the influence of a viscous layer, representing salt strata in nature, embedded within the brittle model materials overlying a VD (Fig. 11d-f) The viscous layer partially decouples the brittle material below it from the brittle material above it during initial extension (Fig. 11e, m). During subsequent compression and inversion, the viscous material will advect along the active faults (Fig. 11f). Similar to the brittle-only base plate models, orthogonal inversion favours the development of new, lower angle thrust faults (Fig. 11f, n), whereas oblique inversion preferentially reactivates the already existing basin boundary faults (Fig. 11i, o).

Adding a basal viscous layer to the base of the set-up, will detach the VD (to a degree) from the cover (Dubois et al. 2002; Konstantinovskaya et al. 2007). Depending on various factors, such as layer thicknesses, extension velocity, viscosity of the detachment layer, and the presence of a single or double VD, a single or double graben system, or even a wide rift zone may develop during extension (compare the rift phases of e.g., Dubois et al. 2002, Konstantinovskaya et al. 2007, and Mattioni et al. 2007, and see Zwaan et al. 2019 for a broader discussion on this topic). In Fig. 11h we show an example with a double graben system developing above the VD, where some viscous material rises below the two rift basins (Dubois et al. 2002; Mattioni et al. 2007). Note that such double grabens can also be formed when applying a narrow patch of viscous material above the VD (Pinto et al. 2010). As this brittle-viscous system is inverted by orthogonal compression, basin boundary faults may slightly reactivate, but the bulk of the deformation is accounted for by new thrust faults rooting at the risen viscous material at the base of the graben, incorporating the original graben into a pop-up structure (Fig. 11i).

Dubois et al. (2002) show that, similar to the models without a viscous basal detachment (Fig. 11j-o), higher degrees of oblique compression during inversion will preferentially reactivate existing normal faults. Such easy reactivation is also observed in Pinto et al. (2010) and Mattioni et al. (2007), even though the latter study involved a complex mechanical stratigraphy with multiple viscous layers. Still, it is not impossible for rift boundary faults to dominantly reactivate in brittle-viscous base plate models undergoing orthogonal inversion (Konstantinovskaya et al. 2007). Even more complex results are obtained by Ustaszewski et al. (2005), who applied a similar set-up involving initial oblique extension, followed by oblique compression. Here the initial oblique extension phase developed en-echelon normal fault structures typical of such a set-up (Tron & Brun 1991; Brun & Tron 1993; Bonini et al. 1997; Munteanu et al. 2014; Zwaan et al. 2021a, b), which were partially reactivated during oblique compression (Fig. 12). Here the rate of compression, and thus the degree of brittle-viscous coupling was shown to have an effect on the resulting inversion structures (Ustaszewski et al. 2005).





As mentioned in section 2.3, Wang et al. (2017) are to our knowledge the only authors to explore inversion of pull-apart
       basins (Fig. 6g-i). However due to the rather specific set-up, including low degrees of transtension and transpression, and the
       sensitivity of such systems to slight deviations from pure strike-slip system (Fedorik et al. 2019), a discussion of the results
       in Wang et al. (2017) is beyond the general scope of this review.


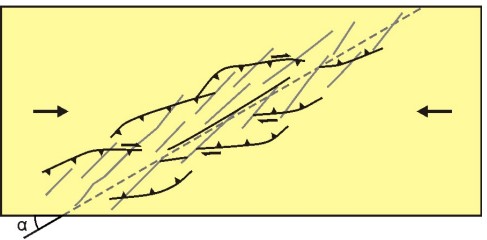

**Figure 12. Top view of an inversion model involving a base plate set-up with brittle-viscous layering and (a) initial oblique
extension leading to a series of en echelon normal faults, followed by (b) subsequent oblique inversion leading to (oblique)**
**thrusting and strike-slip faulting. Inspired by Ustaszewski et al. (2005).**





### 4.4. Distributed basal deformation set-ups

Although a rubber or foam base has been regularly used for the modelling of distributed rifting (Vendeville et al. 1987; McClay et al. 2002, Bahroudi et al., 2003; Bellahsen et al., 2003; Bellahsen and Daniel, 2005; Zwaan et al. 2019), few authors have applied such a basal condition for inversion models (McClay et al. 1989; Amilibia et al. 2005; Dooley & Hudec 2020). The inversion models by McClay et al. (1989) involved a rubber sheet covering the whole base of the model overlain by a brittle layer (Fig. 9a). During extension, the distributed deformation produced a pervasive pattern of normal faulting,

similar to the structures seen in Vendeville et al. (1987) (Fig. 13b). Applying subsequent orthogonal inversion in such a model leads to the development of new thrust faults, crosscutting the previously established normal fault structure that remains inactive (Fig. 13c).

Amilibia et al. (2005) applied an oblique rubber sheet spanned between base plates in their inversion model (Fig. 13a). Even

so, comparing cross-sections at the sidewalls of the model, where the system seems to have behaved in a more or less 2D fashion, with cross-sections from the centre of the model, where deformation was oblique, allows both a 2D and 3D interpretation of this type of model (Fig. 13a-c, j-l). Initial (orthogonal or oblique) extension creates pervasive normal faulting above the rubber sheet (Fig. 13a, j). Subsequent orthogonal compression favours the development of new thrusts rooting along the margins of the interface, whereas the pre-existing normal faults were only mildly inverted (Fig. 13f, k). By

contrast, oblique compression preferentially reactivates the pre-existing (boundary) faults (Fig. 13l).

The study by Dooley & Hudec (2020) aimed at modelling a very specific setting and has a rather complex experimental set-up. The set-up involves offset patches of viscous material overlying a model-wide viscous layer, which itself sits on top of a rubber sheet between two base plates. The model also includes a syn-sedimentary sequence of viscous material representing

salt, inserted between two phases of extension prior to inversion. As such, the 3D results are too intricate for our summarizing purposes in this review paper.

Finally, Richetti et al. (in prep) used foam as an alternative material for inducing distributed deformation in their brittle-viscous models (Fig. 13g-i). In these models a central seed serves to localize deformation during initial extension (Zwaan et

al. 2019), thus creating a graben structure (Fig. 13g, h). Inversion of this structure subsequently leads to the development of thrust faults rooting at the seed, creating a pop-up structure, whereas the initial normal faults remain mostly inactive (Fig. 13i). The authors studied various degrees of oblique extension and compression as well. These results (Fig. 13m-o) are very compatible with the insights derived from the models by Amibilia et al. (2005) (Fig. 13j-l): new thrusting is prevalent during orthogonal inversion, whereas oblique inversion favours reactivation of pre-existing normal faults.






**Overview of inversion models with distributed basal deformation set-ups**

| **Initial set-up (section view)** | **Extension phase** | **Inversion phase** |
|---|---|---|

**Brittle-only I**

a) rubber sheet (below whole model)

b) → c) ←

**Brittle-only II**

d) rubber sheet (between base plates)

e) → f) ←

**Brittle-viscous**

g) seed — foam base (between sidewalls)

h) → i) ←

**Orthogonal vs. oblique inversion** (3D view)

| **After extension** | **Orthogonal inversion** | **Oblique inversion** |
|---|---|---|

**Rubber base, brittle-only**

j) k) α = 0° l) α = 45°

**Foam base, brittle-viscous**

m) n) α = 0° o) α = 45°

brittle pre-tectonic units    syn-rift infill    → applied motion (translation)

viscous pre-tectonic units    post-rift infill    active faults





**Figure 13. Sketches of representative results from brittle-only basin inversion models involving a set-up with distributed deformation at the base. (a-c) Brittle-only inversion experiment with a rubber base below the whole model. (d-f) Brittle-only inversion experiment with a rubber sheet spanned between two base plates. (g-i) Brittle-viscous models with a foam base set-up. (j-l) 3D effect of inversion direction on reactivation of pre-existing faults: oblique compression promotes oblique-slip reactivation of steep normal faults that are unlikely to be reactivated in an orthogonal compression situation. This effect is also observed in the brittle-viscous inversion models with a foam base. Images inspired by McClay (1989), Amilibia et al. (2005), and Richetti et al. (in prep).**

### 4.5. Pre-built basin set-ups

Various authors have used pre-built basins or faults, that were subsequently compressed by moving a backstop into the model for the simulation of basin inversion (Sassi et al. 1993; Panien et al. 2006a; Marques & Nogueira 2008; Di Domenica et al. 2014; Martínez & Cristallini 2017). McClay et al. (2000) used a first phase of differential sedimentary loading to create rift basins to be inverted by a moving backstop. The models by Panien et al. (2005), Del Ventisette et al. (2005, 2006), Sani et al. (2007); Yagupsky et al. (2008), Bonini et al. (2012), Likerman et al. (2013), Munteanu et al. (2014), Jara et al. (2015, 2018), Martínez et al. (2016, 2018) and Granado et al. (2017), who used a base plate to create the initial basin and applied a sidewall for compression and inversion, are very similar in nature to setups with pre-built basins. A general overview of the results from these inversion modelling studies is presented in Figs. 14 and 15.

### 4.5.1. Inversion in section view

The simplest inversion model of this type involves a pre-cut normal fault in a brittle layer (Sassi et al., 1993; Marques & Nogueira 2008) (Fig. 14a). Subsequently, the potential reactivation of this 'fault' (usually a disturbed dilatant zone in granular materials) depends on how much the fault decreases the strength of the brittle layer. A strong fault will not readily reactivate, so that a thrust wedge with newly formed thrust faults will have to accommodate the shortening imposed by the moving backstop, which is in fact very similar to thrust wedge models (Colletta et al. 1991; Cotton & Koyi 2000; Graveleau et al. 2012). By contrast, faults with a shallow dip or weakened with viscous material (Marques & Nogueira 2008) may be reactivated, and a backthrust may develop to form a pop-up structure. Very similar effects are observed in models with a pre-built basin (Fig. 14d-f): a strong basin infill forces the development of new thrusts, whereas a weak basin infill favours reactivation of pre-existing faults, and may even be folded and "squeezed-out" in a somewhat ductile fashion (Panien et al. 2006a; Martínez & Cristallini 2017). Sassi et al. (1993) also showed that the spacing of pre-existing faults influences their reactivation, since not all faults in their models with closely-spaced faults reactivated.





**Overview of inversion models with pre-built basin set-ups**

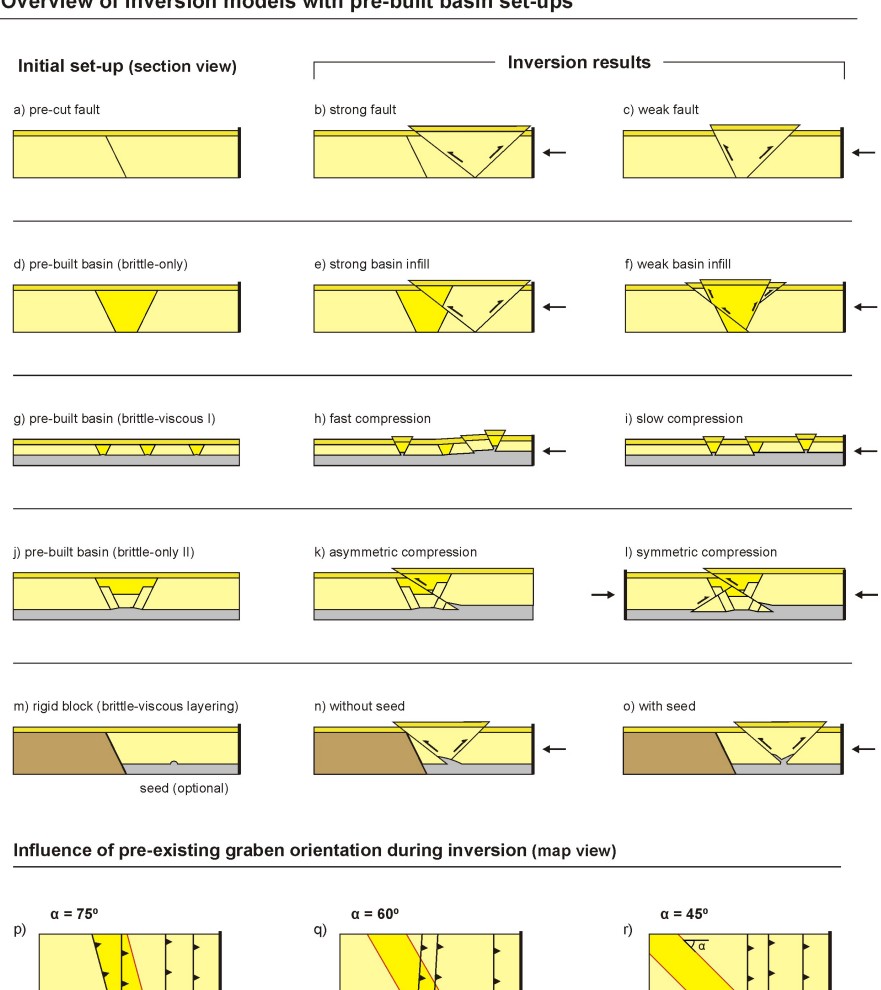

**Influence of pre-existing graben orientation during inversion** (map view)

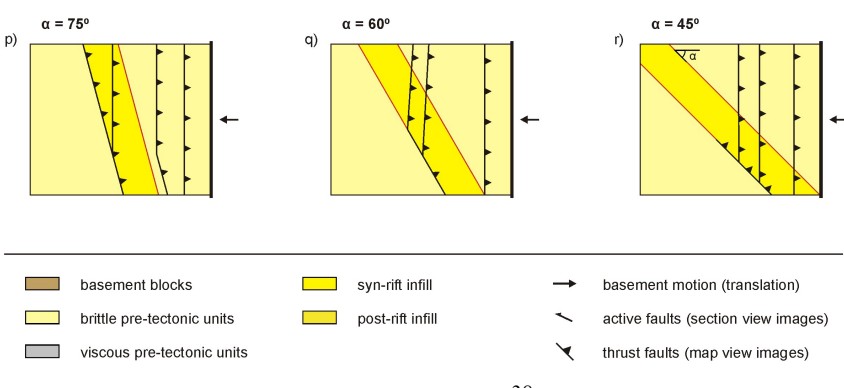





890

895

900

Figure 14. Section-view sketches of representative results from basin inversion models involving a set-up with pre-built basins and faults. (a-c) Inversion of a pre-built fault in a brittle-only layer, which is either relatively (b) strong, or (c) weak. (d-f) Inversion of a pre-built basin, containing either a (e) strong brittle basin infill, or a (f) weak brittle basin infill. (g-i) Inversion of a brittle-viscous system with thin brittle cover layer and dispersed grabens generated by initial extension with a base plate, either during (h) fast compression or (i) slow compression. (j-l) Inversion of a brittle viscous system with a thick brittle cover and a broad graben generated by initial extension using a base plate, either during (k) asymmetric compression, or (l) symmetric compression. (m-o). Inversion of a basin with a viscous basal layer, adjacent to a fixed rigid footwall block, (n) Model without seed, and (o) with seed. (p-r) Effect of pre-existing graben orientation on fault reactivation during subsequent compression and inversion in brittle-only models, shown in map view. Graben orientation is defined as angle α, the acute angle between the graben strike and backstop motion direction. Images inspired by Vially et al. (1994), Letouzey et al. (1995), Panien et al. (2005, 2006a), Marques & Noquera (2008), Yagupsky et al. (2008), Munteanu et al. (2014) and Schori et al. (2021).

The transfer of deformation onto the pre-existing structures away from the moving sidewall in these models is furthermore promoted by the insertion of a detachment layer of microbeads or viscous material (Panien et al. 2006a; Marques & Nogueira 2008; Munteanu et al. 2014). Such a transfer of deformation away from the sidewall is well known from thrust wedge experiments involving a viscous detachment (Colletta et al. 1991; Cotton & Koyi 2000; Borderie et al. 2019; Schori et al. 2021), and is in a way similar to the deformation transfer effect of a base plate during inversion (Fig. 11a-f). Brittle-viscous inversion models by Bonini et al. (2012) illustrate that this transfer of deformation through a viscous layer is affected by the rate of inversion, where deformation during slow inversion is more distributed, whereas deformation during fast inversion is more concentrated towards the moving backstop (Fig. 14g-i). This is evidently caused the strengthening of the viscous material deforming under higher strain rates (Brun 1999). Yet these structures in Bonini et al. (2012) are very different from the overthrusting focussed in the pre-existing basin seen in the models by Munteanu et al. (2014), potentially due to the thicker brittle cover layer in the latter study (Fig. 14j-k). Other variations are likely due to different degrees of extension and compression, as well as the symmetry or asymmetry of compression (Likerman et al. 2013; Munteanu et al. 2014; Jara et al. 2015, 2018, see section 4.5.2).

Vially et al. (1994) and Letouzey et al. (1995) have explored another setup, involving a rigid footwall block next to a pre-made basin containing a brittle infill on top of a viscous layer (Fig. 14m). The authors examined a setup without a seed (Fig. 14n) and one with seeds to simulate the presence of salt diapirs (Fig. 14o). In the situation without a seed, deformation took a shortcut and instead of reactivating the fault along the basement block, the shortening caused the development of a low-angle thrust, along with a backthrust to form a pop-up structure (Fig. 14n). By contrast, the presence of a seed localizes shortening, so that a pop-up structure forms above the seed instead (Fig. 14o). Note that these drawings are simplified versions of the models, and we refer the reader to the original research papers for the detailed model depictions (Vially et al. 1994 and Letouzey et al. 1995).






### 4.5.2. Inversion in 3D

The 3D arrangement and location of pre-existing structures with respect to the shortening direction may strongly affect which of these structures will reactivate and how, as observed in the brittle-only experiments by Panien et al. (2005), Yagupsky et al. (2008), Di Domenica et al. (2014) and Deng et al. (2019) (Fig. 14p-r). These models show that oblique
structures may be reactivated, and similar to (brittle-viscous) Jura Mountain models with oblique basement steps (Schori et al., 2021), the front of each new thrust sheet may partially follow the trend of the pre-existing structures, before reorienting itself to become sub-perpendicular to the general direction of compression (Fig. 14p-r). Further 3D complexities in models with a viscous basal detachment can be induced by having (partial) walls/backstops move inward from one or both sides of the model, after creating an initial basin by means of base plates (Munteanu et al. 2014). As a result, different parts of these
models have undergone different degrees of (asymmetric or symmetric) compression during inversion, leading to significant variations in structural style (Fig. 14k, l).

Jara et al. (2015, 2018) applied rotational motion of a base plate to generate along-strike basin width variations, as well as rotational motion of a backstop for along-strike variations in compression during subsequent inversion. Fig. 15 illustrates an
example of initial rotational extension, followed by orthogonal compression in their brittle-viscous models. The rotational extension leads to the development of a V-shaped basin, with more stretching away from the rotation axis, together with increased tilting of fault blocks (Fig. 15a, cI-III). The development of such V-shaped basins is typical of rotational extension systems (Souriot & Brun 1992; Benes & Scott 1996; Molnar et al. 2017; Zwaan et al. 2020b; Schmid et al., 2021), and the increased tilting of faults with increased amounts of extension is in line with observations from previous brittle-viscous
models (e.g. Mandal & Chattopadhyay 1995; Zwaan et al. 2016). When applying orthogonal inversion, the initial rift geometry has a clear effect on the final model structures (Fig. 15b, cIV-VI). Closer to the original rotation axis, the model is relatively undeformed so that inversion has few weaknesses to reactivate. Therefore, deformation remains relatively close to the backstop (Fig. 15b, c: panel IV), similar to observations in other models (Panien et al. 2005; Bonini et al. 2012) (Fig. 14b, e, g-i). By contrast, farther away from the rotation axis, the more developed graben with its lower-angle normal faults
represents a weakness that is readily reactivated (Fig. 15b, c: panel VI), so that deformation can be transferred farther from the backstop, as also observed by Bonini et al. (2012) and Munteanu et al. (2014) (Fig. 14g-l).





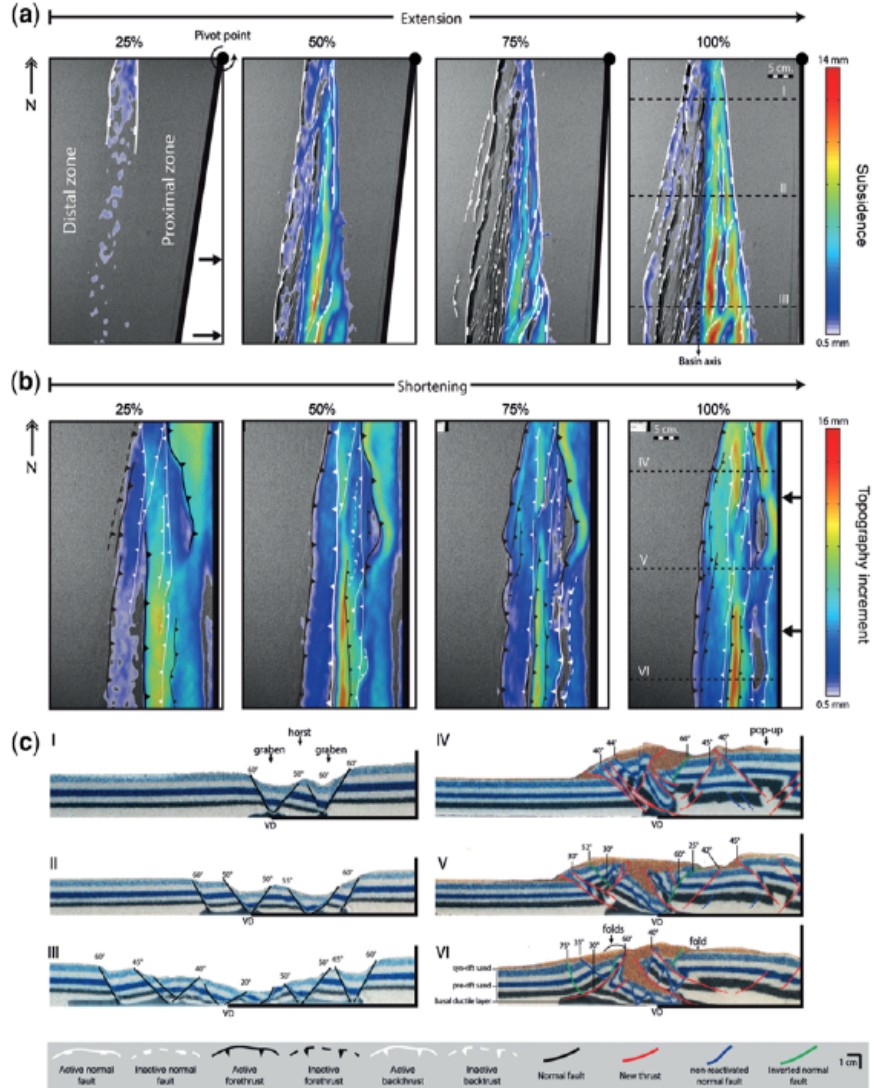

**Figure 15. Rotational inversion model. (a) Extensional phase: active subsidence at different extension percentages. (b)**
**Compressional phase: active topography increment at different shortening percentages. (c) I, II, II: cross-sections at the end of**
**initial extension by means of a base plate set-up. IV, V, VI: cross-sections after homogeneous shortening by means of a backstop**
**moving into the model. Image adopted from Jara et al. (2015), and reproduced with permission from the Geological Society,**
**London.**





### 4.6. Influence of additional geological processes

Even though surface processes are known to affect the development of both extensional and compressional tectonic systems (Koons et al., 1990; Burov & Cloetingh 1997; Buiter et al. 2008; Graveleau et al. 2012; Moragas et al., 2017; Roma et al., 2018; Zwaan et al. 2018a; Borderie et al. 2019), and are thus of importance when considering basin inversion, they have so far received little attention in analogue basin inversion models. Modellers generally apply either full sedimentation or no sedimentation during model runs, with only few studies testing the actual influence of sedimentation on the system. Richetti

et al. (in prep) show how the lack of syn-rift sedimentation causes the basin to be "squeezed" (Fig. 16b), whereas the presence of sediments fills the available accommodation space (i.e. the basin) and strengthens the system, obstructing such "squeezing" (Fig. 16a). Dubois et al. (2002) found that sedimentation prevents the reactivation of some faults in the system, and Pinto et al. (2010) provide similar results, highlighting that added sedimentation during extension and sedimentation reduces fault reactivation. These observations are in line with the results from models involving pre-built basins with a

strong or weak basin infill (section 4.5.1, Fig. 14e, f): in the former case, the strong basin is poorly reactivated, whereas in the latter case, the reduced strength of the weak infill allows reactivation. Extrapolated, the total absence of infill makes the system weaker, increasing the likelihood of reactivation. However, even though applied in a model by Strzerzynsky et al. (2021), the relative influence of erosion has to our knowledge never been systematically tested in basin inversion models.


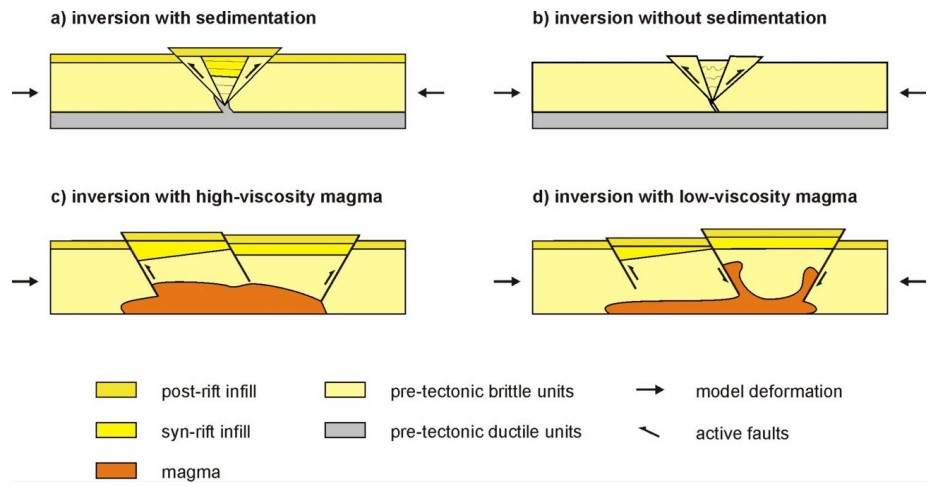

**Fig. 16. (a-b)** Effect of presence and absence of syn-rift sedimentation during rifting (inspired by Richetti et al, in prep). **(c-d) (c-d)** Effect of magmatism and magma viscosity on inversion. Inspired by Martínez et al. (2018).





Similar to sedimentary loading, tectonic loading can have an important influence on basin inversion tectonics. This is most relevant in models with moving sidewalls, which are in fact very similar to thrust wedge experiments (Graveleau et al. 2012, and references therein). Granado et al. (2017) tested the influence of differential loading, revealing that high tectonic loading may prevent fault reactivation, but loading gradients may in fact promote reactivation of sub-thrust basins, i.e., basins below or near the tip of the thrust wedge. These authors have also applied tilting of the basin during inversion, which is known

from critical taper theory and experiments to significantly affect the dynamics of a thrust wedge (Buiter 2012; Graveleau et al. 2012, and references therein).

Apart from the more general impact of (sedimentary) loading, some modellers have also included syn-tectonic deposition of viscous layering in their models, to simulate the accumulation of evaporite (salt) or clay units (Del Ventisette et al. 2006 and

2007; Sani et al. 2007; Mattioni et al. 2007; Dooley & Hudec 2020). As also seen in models including pre-tectonic viscous layering (Nalpas & Brun 1996; Ferrer et al. 2016, Figs. 9d-f, m-o, 11d-f, m-o), these viscous layers act as detachments, causing often very complex deformation (especially when multiple detachments are involved, Mattioni et al. 2007, Dooley & Hudec 2020), as well as diapirism related to the activity of large faults (in a complex interaction with sedimentation patterns, Moregas et al. 2017). Simulated magmatism can have a similar effect, in that it detaches the overlying brittle units from the

model base and can migrate along fault planes (Martínez et al. 2016, 2018). As such, enhanced magmatism during inversion may lubricate faults, and promote the development of pop-up structures with magma-accumulations near the surface, especially when the magma has a low viscosity (Fig. 16c, d).

### 5. Comparison to numerical models and nature

### 5.1. Comparisons between analogue and numerical models

Whereas analogue modellers study tectonic processes by running scaled experiments in the laboratory, other researchers apply numerical modelling methods. These include techniques based on an assembly of particles (Distinct Element Method, DEM) as well as continuum methods (Finite Element Method, FEM). DEM methods may intuitively seem more appropriate for modelling granular materials, but it should be kept in mind that a DEM particle is generally of such size to include many scaled sand particles. Continuum methods do generally not produce discrete fault planes, but have been shown to be well

suited for simulating sand behaviour (Buiter et al. 2006, 2016; Crook et al., 2006). The numerical models allow the inclusion of parameters that are highly challenging to implement in analogue models, such as thermal effects, isostasy, surface processes, parallel deformation mechanisms, and strain weakening. They also readily provide quantitative insights into internal deformation patterns and stress measures that are challenging, if not impossible, to obtain from analogue models. On the other hand, numerical models may lack sufficient resolution when studying tectonic processes in 3D, though this is

rapidly improving, depend on their parametrization of especially brittle processes, and often require access to a high-performance computer cluster. As such, analogue and numerical modelling methods both have their strengths and





weaknesses, and combining these methods for studying tectonic processes can provide more robust results (Ellis et al. 2004; Buiter et al. 2006, 2016; Zwaan et al. 2016; Brune et al. 2017).

Various authors have numerically modelled basin inversion (Hansen & Nielsen 2003; Buiter & Pfiffner, 2003; Buiter et al. 2009; Granado & Ruh 2018; Ruh & Vergés 2019; Ruh 2019), and a number of researchers have applied both analogue and numerical methods, or compared analogue to numerical results. The study by Sassi et al. (1993) elegantly shows how shallow-dipping pre-existing faults are preferentially reactivated in both their analogue and numerical experiments. Buiter and Pfiffner (2003) compare their numerical models of inverted domino faulting to the work by Buchanan & McClay (1992),

finding a fair fit, which validates the results of both modelling studies. Also Panien et al. (2006a) obtained a general good agreement between their analogue and numerical models of a pre-made basin deformed by sidewall compression (Fig. 17), with the models showing similar shear zone structures that highlight the difficulty of reverse reactivation of extensional shear zones in orthogonal shortening. Similarly, Yamada & McClay (2010) found that their numerical models of listric fault basin inversion fit well with their previous analogue modelling study (Yamada & McClay 2004), even though the complex

faulting in the analogue models is not reproduced in the numerical equivalent.

The limited number of studies that use a combination of analogue and numerical modelling techniques to investigate basin inversion may reflect the challenges involved in numerically simulating analogue setups (Buiter et al., 2016). Instead of direct comparisons aimed at achieving similarity in results, we would urge future studies to utilize the strengths of both

methods to investigate complementary processes and factors in basin inversion. For example, analogue models could focus on 3D setups for known material properties, whereas numerical studies could add insights into thermal effects, rheological changes, or surface processes.



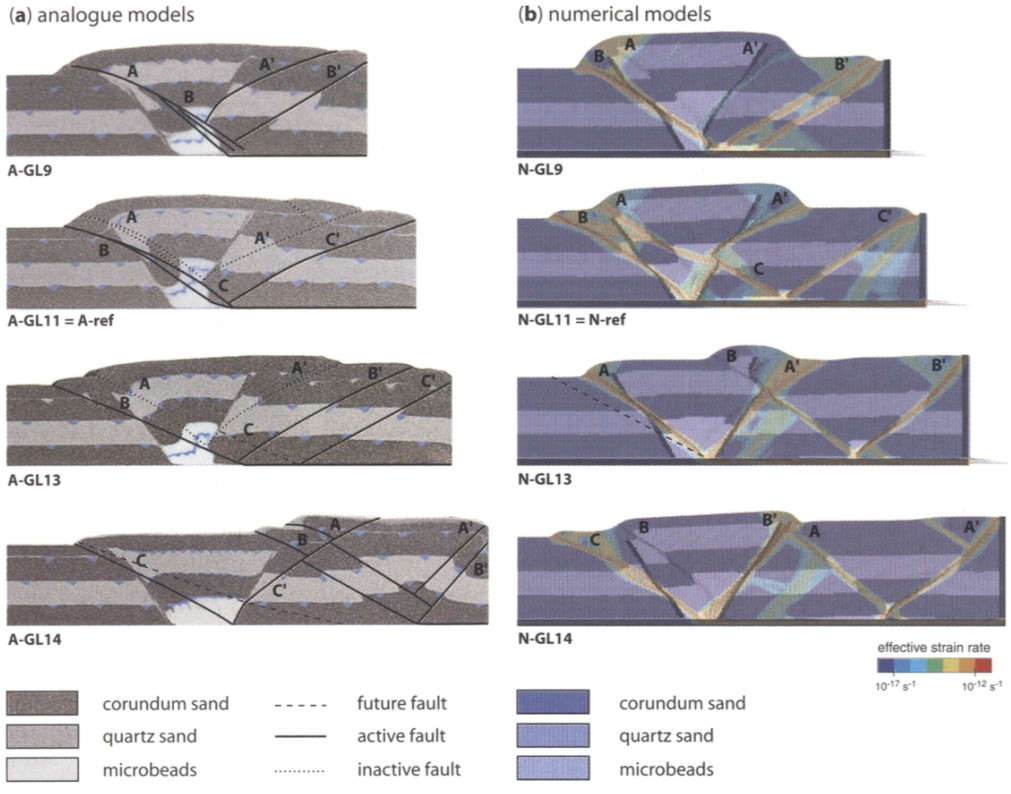

**Figure 17. Example of analogue-numerical model comparison (Panien et al. 2006a). Both set-ups model orthogonal inversion of a pre-built basin. The numerical models are obtained with a finite-element method. Reproduced with permission from the Geological Society, London.**






### 5.2. Comparison of analogue modelling results to natural cases

Here we present two examples published in previous basin inversion modelling studies (Fig. 18). The first comparison concerns a classic example of an inverted basin in the southern North Sea (the Winterton High, Bradley et al. 1989, Figs. 1, 18b), as discussed by Panien et al. (2005). The inversion of the Triassic basin during a series of Cenozoic compression

events that affected large parts of NW Europe (Erratt et al. 1999; De Jager 2003; Evans et al. 2003; Doornenbal & Stevenson 2010; and references therein) expelled the basin infill out of the original graben by reactivating the rift boundary faults. This is very similar to the inversion of the basin infill in the graben set-up used by Koopman et al. 1987 (Figs. 8a-c, 18a). Furthermore, where the faults in the basement forced reactivation of the boundary faults, the propagation of these faults in the weaker post-rift overburden have a shallower dip (Fig. 18b), This is also observed in analogue models (Fig. 18a),

demonstrating the relevance of such analogue model results for our understanding of the dynamic evolution of the Winterton High, and other similarly inverted basins.

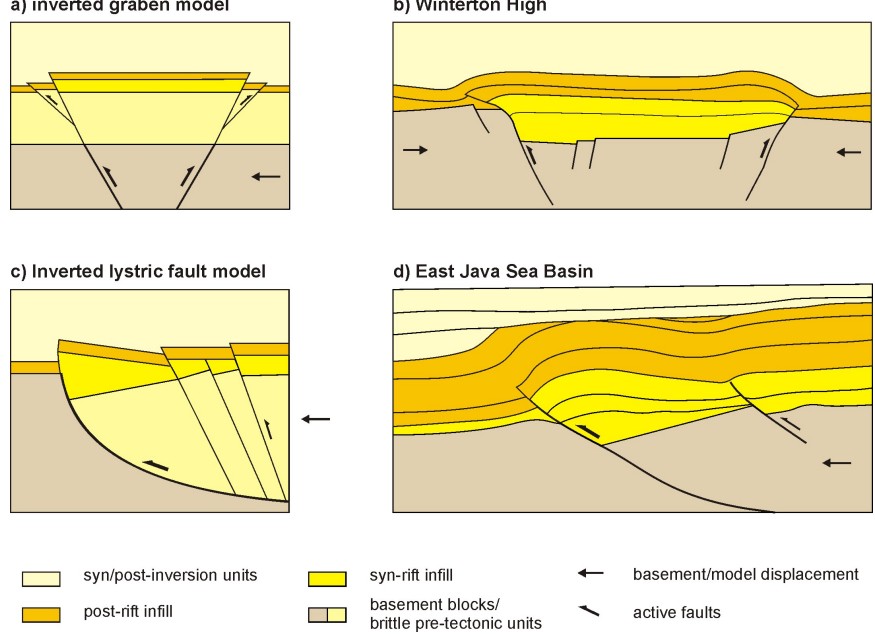

**Figure 18. Comparison of schematic model results with natural examples. (a) Mobile basement graben set-up, see model details in Section 4.1 and Fig. 8c. (b) Winterton High in the southern North Sea (modified after Badley et al. 1989, with permission from the Geological Society, London). (c-d) Inversion of a listric fault basin in (c) an analogue model and (d) in nature (sketch of a seismic line from the East Java Sea Basin by Goudswaard & Jenyon, 1988). Modified after McClay (1995), with permission from the Geological Society, London.**






A second example is provided by McClay (1995) and involves an inverted half graben controlled by a listric fault (Fig. 18c). Both the model and natural example in the East Java Sea (Goudswaard & Jenyon 1988) show very similar structures: thickening of syn-rift sediments towards the listric fault, that are subsequently uplifted as the basin is inverted due to reverse motion along the listric fault. Further similarities are the occurrences of (somewhat) inverted normal faults away from the

main listric fault. Also here, the model nicely fits the natural example, and by examining the model's development over time, we gain valuable insights into the dynamic evolution of the natural example.

Comparing model results with natural examples has indeed provided invaluable insights into basin inversion tectonics. However, one must always keep in mind the limitations of analogue modelling. For instance, most analogue models of basin

inversion do not consider isostatic effects. Isostasy is an important factor in large-scale plate tectonic processes (Burov & Cloetingh 1997), and is therefore included in lithospheric-scale analogue modelling studies of rifting (Vendeville et al. 1987; Nestola et al. 2013, 2015; Molnar et al. 2017; Beniest et al. 2018; Zwaan & Schreurs, 2021, in prep), as well as of collisional and intra-plate tectonics (Willingshofer & Sokoutis 2009; Luth et al. 2010; Sokoutis & Willingshofer 2011; Willingshofer et al. 2013; Calignano et al. 2015; 2017) and basin inversion (Cerca et al. 2005; Gartrell et al. 2005). Yet, as basin inversion is

generally considered to affect basins that did not undergo continental break-up and that are to large degree filled by sediments, the effects of isostasy can be considered limited and the model results valid.

Similar arguments can be made for the influence of magmatism, thermal effects in general, and diagenesis. Even though modelling studies have shown that magmatism can significantly affect basin inversion tectonics (Martínez et al. 2016; 2018),

it is not very common in rift basins prior to break-up, or during subsequent inversion. The same is probably true for thermal effects and diagenesis. Only when the system would go beyond the initial rifting phase, can these effects be expected to become important.

Possibly more impactful limitations are the lack of pore fluid effects in analogue models, which are known to have a

significant influence on inversion tectonics (Sibson 1985, 1995, 2009). Another limitation is the past focus on 2D inversion, and although modelling efforts have more and more explored the third dimension, it is tempting to think of basin inversion as a 2D process (especially given the legacy of 2D seismic sections). Ongoing developments in the analogue modelling community, as well as in the world of seismic acquisition, are however making the analysis of 3D modelling results much more straightforward (see also section 7).


Still, the comparisons presented above highlight the crucial use of analogue models for simulating the dynamic development of tectonic systems that take millions of years to unfold. This is especially true when combining these analogue models with numerical modelling techniques, as discussed in the preceding section.



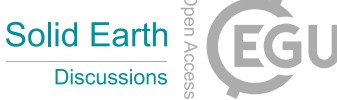

## 6. Synopsis of main insights

The model results described above give insight into the processes and factors that result in successful basin inversion or may prevent a basin from being inverted.

The first requirement for successful (basin) inversion is the presence of a basin or fault structure that provides a relative weakness relative to the immediate surroundings to focus compressional deformation. Such weakening can originate from

the infill of the basin, where weak sediments facilitate inversion (either by allowing fault reactivation, the nucleation of new faults near the basin edge, or upward folding of the infill). By contrast, a strong infill may prevent inversion (Panien et al. 2006a Fig. 14d-f). Similarly, normal faults that are either weakened or have a low dip angle are more likely to be reactivated (Marques & Nogueira 2008, Fig. 14a-c), which is in line with fault theory, which otherwise predicts that new thrusts should form (Section 2). The preference for forming new thrust faults is particularly well known from the propagation of inverted

normal faults into the post-rift overburden in models with (mobile) basement blocks (Fig. 8c, f, o). Finally, the shape of pre-existing normal faults (e.g. straight, listric or undulating) can strongly affect the resulting inversion structure as well (McClay 1996, Figs. 9, 10).

A second requirement for inversion is the transfer of deformation into the basin or fault structure. Apart from the relative

weakness of the modelled basin or fault, this depends on the boundary conditions, i.e., the model set-up, as well as the mechanical layering used in the model. Model set-ups involving deformation driven from the base are more likely to efficiently induce inversion. By contrast, inversion driven by a backstop generally needs a detachment layer of some sorts (either microbeads or a layer of viscous material) that decouples the overburden from the base. Otherwise, deformation will simply develop directly in front of the backstop (Fig. 14). In this context, a base plate set-up in fact acts as a sort of

detachment as well, efficiently transferring deformation deep into the model (Fig. 9). Weak layers within the model materials can also act as detachments, leading to new levels of complexity (Nalpas et al. 1996; Di Mattioni 2007; Ferrer et al. 2016; Dooley & Hudec 2020; Figs. 8-16).

A third factor favouring inversion is oblique shortening. Whereas high-angle normal faults may be often (partially) ignored

during orthogonal inversion in favour of newly formed thrust faults, oblique inversion can readily reactivate these normal faults. Such reactivation of normal faults during oblique inversion is well-documented in analogue models with a variety of different set-ups (Figs. 11j-o, 13j-o), and is explained by the reduced fault angle with respect to the orientation of the principal stresses (see fault theory specified in section 2). Moreover, along-strike variations in the models due to variations in basement block or base plate geometries, differently oriented pre-existing structures and weak zones, or rotational extension

or compression can cause highly complex distributions of structures (Figs. 10p, 12s, t, 14p-r, 15). These complexities highlight the importance of considering the third dimension when studying tectonic processes.



Finally, additional geological processes such as sedimentation and magmatism can affect inversion processes as well. Sedimentation and erosion change the normal stress in the model layers, thus changing the frictional strength and affecting

the likelihood of inversion. Magmatism can facilitate fault reactivation when intruding along fault planes, thus reducing fault strength (Fig. 16).

The insights from these analogue models are in line with the mechanisms of basin inversion summarized in section 2, and generally concur to a high degree with results from numerical modelling studies, validating both modelling methods (Fig.

17). Furthermore, these insights have proven highly valuable to better understand the evolution of basin inversion tectonics in natural settings (Fig. 18).

**7. Perspectives for future analogue modelling studies of basin inversion**

In this review we describe the current state-of-the-art of analogue modelling of basin inversion. There are however many opportunities for improvements and future modelling studies, revolving around new modelling methods, the use of improved

model analysis techniques and the combination of analogue and numerical modelling methods, benchmarking studies, and establishing best practices in analogue modelling.

**7.1.  New modelling methods to tackle new questions**

Improvements in experimental techniques can include the development and application of new model materials (e.g.

Boutelier & Oncken 2011; Boutelier et al. 2012; Abdelmalak et al, 2015; Zwaan et al. 2016, 2018a; Mayolle et al. 2021; Massaro et al. 2021). Such new materials allow the implementation of different degrees of viscosity in viscous materials (Zwaan et al. 2018b), or modifications to the cohesion of brittle materials (Abdalmalek et al, 2015; Montanari et al. 2017; Massaro et al. 2021). Further opportunities lie in the application of new materials with elastic (-plastic) behaviour such as gelatin for the study of inversion-related seismicity (Rosenau et al. 2009; Di Giuseppe et al. 2009), or visco-plastic behaviour

such as kinetic sand for better simulating the brittle-ductile transition (Mayolle et al. 2021), or temperature effects (e.g. Boutelier & Oncken 2011; Boutelier et al. 2012).

Other improvements are related to the development of new modelling apparatus that allow for improved simulation of tectonic processes and new model boundary conditions (e.g., Molnar et al., 2017; Zwaan et al. 2020b; Zwaan & Schreurs

2021, in prep, Eisermann et al. 2021). An interesting development is the application of force boundary conditions, rather than velocity boundary conditions (Gartrell et al. 2005; Konstantinovskaya et al. 2007). Constant force during rifting may explain rapid changes in tectonic deformation rates, for instance when the strength of the tectonic system is reduced due to



necking of the lithosphere (Brune et al. 2016). Since increasing strain rates increase the degree of brittle-viscous coupling in tectonic systems (Brun 1999; Bonini et al. 2012; Zwaan et al. 2021a, b), such increases (and decreases) in strain rate can

significantly change the style of deformation during basin inversion as well (Bonini et al. 2012), and could be further explored, for instance using models in which deformation rates are programmed to be variable.

The majority of basin inversion models have been conducted on an (upper) crustal scale, motivated by the search for hydrocarbon reserves (e.g. in the North Sea). The insights gained from these models can also be of use for $CO_2$ sequestration

and mineral exploration. In addition, part of the future of basin inversion modelling may lie in the study of mantle exhumation as hyperextended basins are inverted. The serpentinization of mantle rocks that were initially exhumed in hyperextended rift basins, and are now incorporated in mountain belts, may be a source of future natural hydrogen production (Dumagin 2019; Lefeuvre et al. 2021). The broad presence of mantle rocks in mountain belts, both in Europe (Faul et al. 2014; Frasca et al. 2016; Schmid et al. 2017) and elsewhere (Vaughan & Scarrow 2003; Dilek & Furnes 2011)

indicates some highly interesting potential. Given the need for truly clean energy production in our efforts to realize a sustainable economy (Gaucher 2020; Moretti & Webber 2021; Scott 2021), researching the tectonic processes causing and controlling basin inversion on a lithospheric scale through novel analogue modelling techniques is more relevant than ever.

Linked to such large-scale inversion processes, is the influence of tectonic loading, which has not received much attention in

analogue modelling studies. Yet, the presence of a thick orogenic wedge covering a basin will have important consequences for inversion processes (Granado et al. 2012; Kiss et al. 2020; Musso Piantelli et al. in prep, Fig. 19). Another research topic to pursue is the effect of magma intrusion during inversion, pioneered by Martínez et al. (2016; 2018). Moreover, the healing of faults over the period between initial extension and subsequent compression could cause important differences in fault reactivation (Hunfeld et al. 2020; Rudolf et al. 2021). Further attention could be dedicated to the inclusion of the interactions

between tectonics and surface processes (Graveleau & Dominguez 2008; Graveleau et al. 2011, 2015; Reitano et al. 2020, 2022; Strzerzynsky et la. 2021), which may also significantly affect inversion processes (section 4.6). In addition, 3D aspects of inversion can be a fruitful avenue for further model studies, especially since basin inversion has traditionally been modelled from a pseudo-2D point of view. Various modellers have indeed started to study complex 3D inversion tectonics, but a whole new field of play, so far to our knowledge only explored by Wang et al. (2017), is the inversion of pull-apart

basins. Finally, changing (fluid) pressure has been simulated in analogue models using air flow systems (Cobbold et al. 2001; Mougues & Cobbold 2003), which could perhaps serve to reproduce changes in pore fluid pressure that are known to have important effects during inversion (Sibson 1985; 1995, 2009).



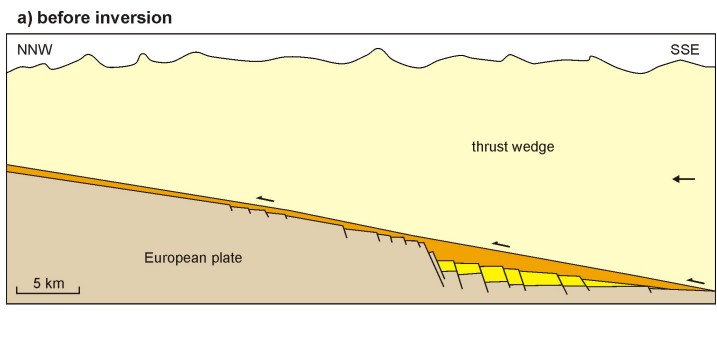

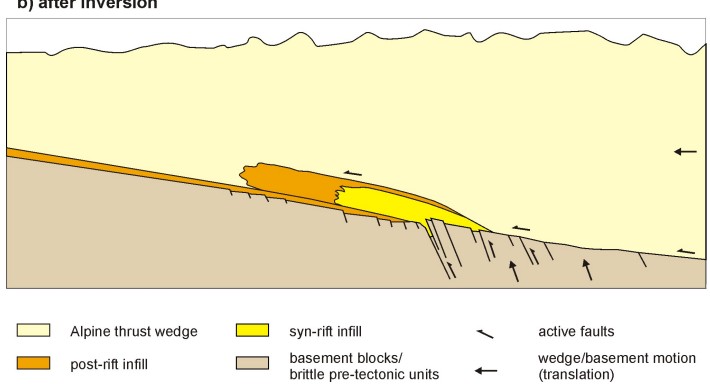


**Figure 19. Inversion of the Doldenhorn Basin in the Swiss Alps, which was once a basin on the European passive margin of the Piemont-Ligurian Ocean, under a >10 km thick thrust wedge. The burial depth causes ductile, rather than brittle behaviour during inversion. Modified after Musso Piantelli et al. (in prep).**

**7.2. Improved (applications of) model analysis techniques**

In parallel with the ongoing development of modelling methods, techniques to analyze analogue models have significantly improved as well, marking the general shift in the analogue community from producing qualitative model results to quantification of these results. So far, various analysis techniques have only been sparsely applied to monitor and analyze basin inversion models. We propose that all tectonic modelling laboratories strive to develop the capabilities for combined

use of DIC techniques, topography analysis and systematic cross-sectioning.



DIC and topography analysis have indeed become standard techniques in the analogue modelling community over the past years. New developments not only provide detailed insights into displacements and strain (Schmid et al. 2021; Zwaan et al. 2021a, b), but also allow an analysis of the type of faulting (normal, reverse or strike-slip, Broerse et al. 2021; Krstekanić et al. 2021). However, DIC methods have so far only seen very limited use in basin inversion models (Wang et al. 2017; Dooley & Hudec 2020; Richetti et al. in prep). Similarly, topography analysis, although routinely applied in tectonic laboratories (section 3.6.3), has been relatively seldomly featured in basin inversion studies. Therefore, we urge basin inversion modellers to adopt these powerful analysis techniques in future studies.

Cross-sectioning has been widely used in inversion modelling works (section 2.6.2). Wherever possible, these cross-sections could be used to create 3D representations of the internal model structural architecture (Ferrer et al. 2016, Roma et al. 2018a, b, Dooley & Hudec 2020, Fig. 7d). Perhaps even better is the use of CT-scanning for analogue model analysis, which has been applied to a surprising degree in basin inversion studies. Next to the detailed 3D analysis of the evolution of model-internal structures (Konstantinovskaya et al. 2007, Chauvin et al. 2018, Fedorik et al. 2019; Lathrop et al. in prep), such CT data also allow for advanced 3D DIC analysis, or Digital Volume Correlation (DVC), yielding unique quantified constraints on model-internal displacements in 3D (Adam et al. 2013; Zwaan et al. 2018a; Poppe et al. 2019).

Other (novel) methods that have been recently applied to the analysis of analogue models, and which would be highly interesting for basin inversion models as well, are the measurement of local stresses in the model by means of stress-sensitive beads (Daniels et al. 2017; Ladd & Reber 2020). It would also be possible to monitor stress through sensors on the model sides (Reber et al. 2014; Herbert et al. 2015; Ritter et al. 2018a, b), or even within the model (Nieuwland et al. 2000; Moulas et al. 2019). Finally, recent workers have pioneered how the reorientation of initially randomly distributed magnetic particles may reveal strain in models (Almqvist & Koyi 2018; Schöfisch et al. 2021). As new analysis techniques are being developed in the analogue modelling community, these could be readily applied to basin inversion studies. A powerful means to contact experts, set up collaborations and gain access to new methods, or to share analogue modelling knowledge in general, is the EU-funded EPOS Multi-scale laboratories network (https://www.epos-eu.org/tcs/multi-scale-laboratories).

Considerable improvements can be made by routinely integrating analogue modelling efforts with numerical studies and techniques. Not only does the combined use of analogue and numerical modelling provide more robust results (section 6.2), the scripts and algorithms used to process and quantify deformation in numerical models can often be used for the analysis of (quantified) data derived from analogue models. An intriguing opportunity would be the application of machine learning techniques (Corbi et al. 2019) for the semi-automatic analysis of analogue models. By doing so, modellers can truly bring together the best of both modelling worlds.





### 7.3. Benchmarking

Directly comparing the results from different analogue modelling studies is often challenging, due to the many small and larger differences in model set-up, materials and applied boundary conditions. And even if researchers aim to run the exact same model, some differences between the model results are to be expected due to variations in handling techniques or

laboratory conditions (Krantz 1991b; Lohrmann et al. 2003; Schreurs et al. 2006; 2016; Maillot 2013; Klinkmüller et al. 2016; Rudolf et al. 2016; Schmid et al. 2020). The overviews presented in this review should therefore be taken as a guide, and the reader is referred to the specific studies for more details on the original model results.

Nevertheless, in order to distinctively determine the relative importance of set-ups, materials and model parameters,

systematic comparisons and benchmarking efforts, such as those done for thrust wedges and rifting (Schreurs et al. 2006; 2016; Zwaan et al. 2019), are needed. Within the field of analogue modelling of basin inversion, there is a great need for benchmarking, as virtually no such efforts have been published. Instead, most studies are focussed on the study of a specific set-up, or aim to unravel the structural history of a natural example. The generalized overviews presented in this review may count as a first stimulus for such a benchmark, but need to be supported by systematic modelling efforts with standardized

set-ups and methods.

Benchmarking efforts may also allow to fill in the various knowledge gaps, as various combinations of model set-ups and materials have not been tested. Rerunning experiments in a systematic manner will also provide an opportunity for detailed analysis with state-of-the art methods (see section 7.3). Such improved analysis will yield a wealth of data and new insights

that would otherwise remain unnoticed (compare e.g. Zwaan et al. 2020a with Schmid et al. 2021).

### 7.4. Best practices

Finally, we would like to point out the responsibility of researchers to describe their research efforts as detailed as possible. This includes the description of the model set-up, experimental materials and model preparation details, as well as an

extensive and systematic description of the original model results (see summary Table A1 in the Appendix). Such detailed descriptions are crucial to ensure model reproducibility and to allow the reuse of established methods for future modelling studies. Indeed, differences in handling methods and lab conditions can considerably affect model results (see section 7.3). Therefore, descriptions of methods and lab procedures should be as extensive and standardized as possible. Describing these details is part of the general shift in analogue modelling over the years, from a qualitative research method to a gradually

more systematic, quantitative science as the techniques and methods steadily improved (Koyi 1997; Ranalli 2001; Graveleau et al. 2012; Schellart & Strak 2016; Zwaan & Schreurs in press). It is clear that not all details and results can always be included in the 20 or so pages of a scientific publication, but these can be published as supplementary materials, often in





digital form, in parallel to the main scientific articles. Such supplementary material can consist of written descriptions, schematic representations, systematic overviews as well as photos and videos, and may be stored on the publication

webpage, or could be stored in independent repositories. These repositories should be organized according to the FAIR principles, so that the data will be openly available to the community (e.g., GFZ Data Services, https://dataservices.gfz-potsdam.de has been routinely used for the storage of modelling results by members of the EPOS Multi-scale laboratories network). This would ideally be accompanied by publishing the main article under an Open Access licence, making the research publicly available.

**8. Concluding remarks**

Basin inversion is a great research topic for analogue modelling studies, and a thorough understanding of the processes involved is of great importance to both science and society. In this review we provide an up-to-date summary of the state of analogue modelling of basin inversion processes. In addition to reviewing the past modelling efforts we also shed light on future modelling challenges and identify a number of opportunities for follow-up research. It follows that basin inversion

modelling can continue to bring valuable new insights, providing a great incentive to continue our efforts in this field. We therefore hope that this review paper will form an inspiration for future analogue modelling studies of basin inversion.

**Acknowledgements**

The Swiss National Science Foundation is acknowledged for providing funding for this research (grant 200021-178731, http://p3.snf.ch/Project-178731), and for covering the Open Access publication fees. M. Rudolf has been funded by Deutsche

Forschungsgemeinschaft (DFG) through grant number 235221301 - CRC 1114 "Scaling Cascades in Complex Systems", Project B01 "Fault networks and scaling properties of deformation accumulation." Research of O. Ferrer has been supported by the SABREM research project (PID2020-117598GB-I00), funded by MCIN/ AEI /10.13039/501100011033. The idea of the review paper was sparked during an EPOS, multi-scale laboratories (MSL) community meeting, and we thank EPOS MSL (https://www.epos-eu.org/tcs/multi-scale-laboratories) for providing an excellent discussion platform for such

collaborative efforts.




## Appendix

**Table A1: Check-list of parameters to record for analogue models of tectonic processes**

| | Information be described / provided | Comments |
|---|---|---|
| **1. Lab conditions** | Temperature<br>Humidity | |
| **2. Model materials** | | |
| *2a. Brittle materials* | Material: bulk composition, mixture components<br>Grain size: range and distribution<br>For clay: water content<br>Density (in model)<br><br>Internal friction (peak, stable and reactivation)<br>Cohesion | Specify the producer of the material(s)<br><br><br>Depends on handling method (e.g. sieving, pouring, scraping), which needs to be specified |
| *2b. Viscous materials* | Material: bulk composition, mixture components<br>Density<br>Viscosity<br>Rheology (Newtonian, Power law)<br>Temperature-dependency of rheology | Specify the producer of the material(s)<br><br>May depend on model strain rate<br>May depend on model strain rate |
| **3. Model set-up and preparation** | Deformation mechanism<br><br>Model dimensions<br>Model layering and layer thicknesses<br>2D and 3D variations in the model<br>Sidewall and basal friction<br>Methods to reduce boundary effects<br>Scaling: parameters and calculations | Base plate set-up, backstop, etc.<br><br><br><br>Lateral variations, seeds, pre-cut faults, etc.<br><br>Lubrication along sidewalls, etc. |
| **4. Model run** | Deformation: velocity, direction, duration<br><br>Application of surface processes: type of sedimentary infill (material), methods of application, sedimentation/erosion intervals | Define any changes (in case of multiphase deformation models) including time between changes |
| **5. Monitoring and analysis** | (3D) photography: camera type, resolution, time-lapse intervals, number of cameras and orientation (top view or oblique view)<br>Cross-sections: method, locations, spacing and orientation<br>DIC: 2D or 3D, analysis interval, resolution, software<br>Topography: method (laser scanning, photogrammetry), analysis interval, resolution, software<br>CT-scanning: device, scanning intervals, resolution, software | In case of 3D "seismic" analysis: software |
| **6. Results** | Systematic overviews of model results: Top views, interpreted sections, topography maps, topography profiles, strain and displacement analysis (DIC), 3D internal analysis, CT data.<br>Videos of model results: top view time lapse, topography, CT-imagery etc.<br>Description of boundary effects<br>Description of failed models | Either in main publication, or supplementary materials (e.g. a data publication with doi) |



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
