# Peer review of "Analogue modelling of basin inversion: a review and future perspectives"

_Solid Earth, 2022_

## Author Response (AR1)

Solid Earth Discuss., author comment AC1
https://doi.org/10.5194/se-2022-8-AC1, 2022

[Figure]

**Reply on RC1**

Frank Zwaan et al.
* * *
Author comment on "Analogue modelling of basin inversion: a review and future perspectives" by Frank Zwaan et al., Solid Earth Discuss., https://doi.org/10.5194/se-2022-8-AC1, 2022
* * *
**Review of "Analogue modelling of basin inversion: a review and future perspectives".**

Authors: Frank Zwaan, Guido Schreurs, Susanne J.H. Buiter, Oriol Ferrer, Riccardo Reitano, Michael Rudolf and Ernst Willingshofer.

**Overview and recommendation**

This article is a review of analogue modelling studies of basin inversion processes. The introduction is well organised and well written. The authors give a clear definition of the process of basin inversion in the context of their review. The aim of this review is clear, and the outline of the article is presented.

In a first part, the authors give an up-to-date state of the art about basin inversions processes through analogue modelling studies. They present the mechanics of basin inversion, analogue modelling techniques, as well as detailed setups examples and typical scaling parameters. Representative results of basin inversion are presented based on their setup. Insights about the governing parameters of basin inversion are given.

The authors then compared analogue and numerical models of basin inversions. I think it is a good idea to make such comparison. I understand it is not the scope of the publication, but this part could be improved a bit by giving more detailed examples. By comparison to natural examples, the authors show the limitation of analogue modelling of basin inversion. To finish, the authors give perspectives and recommendations for future analogue modelling studies of basin inversion.

I think that the subject of this article is interesting and very attractive for the readers of Solid Earth.

The authors provide an impressive overview of what has been done and what should be improved in future analogue modelling studies of basin inversion. The overall manuscript very well organized, well written, and well illustrated. I recommend **accepting this**

**article with really minor revisions**.

- **Author's reply:** we thank the reviewer for these kind words and the positive assessment of our submitted manuscript. We have processed the minor comments in our revised manuscript.

**Minor comments**

Line 138: The title of part 2 is missing (before 2.1 Mechanics of basin inversion).

- **Author's reply:** The title should be "2. Mechanics of basin inversion" instead. We have corrected this.

The authors did not mention anything about edge effects along the sidewalls of the model setups. Could there be such effects in the different initial conditions presented in the study? If yes, how to prevent edge effects or how to characterize it?

- **Author's reply:** We agree that boundary effects are an important factor in analogue models. We already mentioned the potential influence of sidewalls/glass boundaries on model results (section 3.6.1 on photography [model monitoring]). We also mentioned the importance of model preparation and handling methods in section 4 (first paragraph), and this also comes back in discussion sections 5.2 and 7.1). As pointed out at the start of section 4, there is a wide range of set-ups, so that we resort to simplified overviews. As such, there is not proper occasion to go too much into detail regarding boundary effects (also because the text is already rather long). We have however added some details to the first paragraph of section 4 to point out the importance of boundary effects (incl. Some references, e.g. works by McClay & co-authors, Schreurs et al. 2006, and Souloumiac et al. 2012), and we hope that this will suffice.

Figure 15 has a low resolution. Please improve it.

- **Author's reply:** The figure in the manuscript is (close) to the original resolution of the figure, as published in Jara et al. (2015). We have now inserted the highest possible resolution figure, and replaced the less readable text and annotation with high-resolution text/annotation.

Caër et al., (2015) provide a parametric analysis of the reactivation of a normal fault through numerical modelling using Limit Analysis. I think Limit Analysis could be mentioned as a numerical modelling technique. This methodology requires few inputs parameter. As such, it is in a way close to analogue modelling and could be easily compared to it.

- **Author's reply:** We thank the reviewer for this suggestion and have now cited the paper in section 5.2 (about linking numerical methods to analogue models)

Caër, T., Maillot, B., Souloumiac, P., Leturmy, P., Frizon de Lamotte, D., Nussbaum, C., 2015. Mechanical validation of balanced cross-sections: The case of the Mont Terri anticline at the Jura front (NW Switzerland). J. Struct. Geol. 75, 32–48. https://doi.org/10.1016/j.jsg.2015.03.009

**Citation**: https://doi.org/10.5194/se-2022-8-RC1

[Figure]

Solid Earth Discuss., author comment AC2
https://doi.org/10.5194/se-2022-8-AC2, 2022

[Figure]

**Reply on RC2**

Frank Zwaan et al.
* * *
Author comment on "Analogue modelling of basin inversion: a review and future perspectives" by Frank Zwaan et al., Solid Earth Discuss., https://doi.org/10.5194/se-2022-8-AC2, 2022
* * *
**GENERAL COMMENTS:**

I enjoyed reading the manuscript very much and especially reviewing the figures.

The figures are the heart of the review. They are easy to follow and show a very good comparison to previous studies. There is a careful design of the figures and a careful explanation in the 'figure captions'.

This work will be a great contribution to all modelers. It is an update on analysis in general and for those of us who work on basin inversion analysis. Undoubtedly, this work will be a guide and excellent analog modeling benchmarking for the field of basin tectonic inversion.

The number of works reviewed is impressive; I counted 285 publications in the reference list. Amazing. Pay attention to cross-checking citations and references. See specific comments below. Being this manuscript is a review, it is even more important that there are no doubts about the reference to the reviewed works so that the reader can find them without problems in case they are interested in delving into one of the topics.

In addition, a careful review of the published models is noted, with care in the experimental conditions, which is shown in the explanations and analysis of the models within the text.

This is reflected in a very well-written document, generally clear, step by step, with adequate detail.

In the last item of the 'review' 'perspectives for future modeling studies of basin inversion', the authors take a good look at the analog projection in the field of tectonic inversion of basins. The authors give good recommendations to the modelers to homogenize the experimental conditions in the future, to facilitate comparison and analysis of the results. It seems to be a very good closing of the review.

All in all, excellent work.

Below I send specific comments and technical corrections on the revision made.

- **Author's reply:** we thank the reviewer for these kind words, and the positive assessment of our submitted manuscript, as well as the constructive feedback. We have processed the various comments in our revised manuscript (see details below).

**SPECIFIC COMMENTS:**

The schematization of figures 6, 8, 9, 11, and 14 stand out.

Table 1 presents a good summary of the experimental criteria analyzed.

- **Author's reply:** we thank the reviewer for these kind words

Sometimes there is an abuse of using quotes without clearly explaining the ideas that are to be discussed in the paragraph.

- **Author's reply:** We thank the reviewer for pointing out these issues and have solved them as specified below. Furthermore, we have double-checked the manuscript for possible further issues, and have corrected these wherever we deemed necessary. A thorough final check will be done before resubmission.

For example, in item 1.2. mentions many references but does not specifically explain the importance of 'basin inversion tectonics'.

- **Author's reply:** We have added some specifications in the revised manuscript.

In L. 91 authors comment that "determining timing of investment has been historically crucial to petroleum geologist". Authors should state directly why it is 'crucial', without leaving it to the imagination of the reader.

- **Author's reply:** We have now added some more details: inversion cannot only create traps for hydrocarbons, but can also shut down hydrocarbon production by uplifting/exhuming the deeper parts of basins (which are poorly resolved on seismic data). As such, it is crucial to petroleum geologists to understand basin inversion.

In L. 96 they indicate '… development of mineral resources'. This is very broad. What mineral resources'' are they referring to?

- **Author's reply:** The literature we use to discuss mineral resources covers ore deposits and economic minerals (i.e., Pb-Z mineralizations, Fe-Cu-Au deposits, etc.). We now specify this in the text.

Idem L.97-98 indicates that 'understanding of basin evolution, including basin inversion, is

also of great interest for geothermal energy projects'. Authors should state directly why it is 'crucial'.

- **Author's reply:** We have now added the specification that tectonic motion can significantly affect the thermal profile of the subsurface, and as such is important to understand for geothermal energy projects.

Another example is the paragraph between L. 901 and 913; it is very difficult to follow. Only works are cited. For readers who do not know them, they would have to review them to fully understand the ideas that are being proposed in the paragraph. I suggest being more explicit.

- **Author's reply:** We believe that this paragraph is rather clear. What was perhaps missing was an early reference to Fig. 14g-i, which is now included, plus a reference to compare Fig. 14g-i to Fig. j-k (the latter of which has a thicker viscous basal layer).

Idem in the paragraph between L. 980 and 986. At least mention some figure that allows following the ideas.

- **Author's reply:** Similar to the previous comments, we think that the paragraph is rather clear. We decided to not include a summary figure due to the relative complexities of Granado et al. (2017) models that are discussed here, and have added a clear reference to check the original paper for more details.

**TECHNICAL CORRECTIONS:**

**Text.**

At the end of item 1.3 you should cite a reference work, L. 128

- **Author's reply:** We have added citations to Koyi (1997), Ranalli (2001), Bonini et al. (2012), and Graveleau et al. (2012), who adequately present the evolution of analogue modelling techniques over the past 150 years or so. Also Hubbert (1937) is now mentioned earlier on in the paragraphs (with Hall 1815 and Cadell 1889)

Homogenize the style of citations. Sometimes the authors use '&' and sometimes 'and' to cite a work by 2 authors.

- **Author's reply:** We thank the reviewer for noticing these inconsistencies. We have corrected things and replaced the instances of "&" with "and" (in accordance with the Solid Earth guidelines for in-text references).

In some parts of the text, there are leftover commas. Example L.230

- **Author's reply:** We thank the reviewer for noticing these flaws. We have gone through the text and removed where these were left over (and inserted them, where missing).

A thorough final check will be done before resubmission.

Dots are missing, for example L. 680, L. 782, L. 1049.

- **Author's reply:** We have gone through the text and added periods where these were missing (and removed them where they should not be). A thorough final check will be done before resubmission.

Leftover an 'and' L. 791

- **Author's reply:** In fact, the "and" is correct here, as the first three citations are grouped, followed by the second part of the sentences. But we see how it can be confusing to the reader, and have replaced it with "or" now.

L. 1033 says 'Tecotnics', it should say 'Tectonics'.

- **Author's reply:** We believe that the reviewer means line 1435 here, where the journal "Tectonics" was indeed misspelled. It is corrected and we thank the reviewer for spotting the typo.

Find an alternative to the word 'relative' in lines L. 1098-1099.

- **Author's reply:** we have replaced the second "relative" with "compared"

Review cut words 'exhumation' L. 1615, or 'modeling' L. 2024, 'coefficients' L. 2050, 'modelling' L. 2183

- **Author's reply:** Thanks for spotting these typos. We have corrected these and others instances of hyphenation after a thorough check of the manuscript.

L. 2165: Says 'Geoterhrmics', should say 'Geothermics'

- **Author's reply:** It is corrected.

Figures:

The yellow colors of syn-rift and post-rift hardly differ. Try to distinguish them. Figs 8, 9, 11, and similar ones.

- **Author's reply:** We agree that the figure colors can be improved. We are experimenting with combinations so that we can resubmit figures with an optimal clarity.

Fig. 2. The sigma symbols are very small, in printed format, and they are barely visible. Increase size.

- **Author's reply:** We agree that these symbols (i.e. the subscript parts of the sigma symbols) are too small and have increased the font size in the new version.

Fig. 9. Correct in letter g) is 'Listric' not 'ramp-flat-ramp'

- **Author's reply:** thanks for spotting this error, we have corrected the typo.

Fig. 10. The colors of the surfaces of the models are confusing, they should be some color other than the yellow palette. Also, orange colors do not appear in the legend.

- **Author's reply:** We have corrected these issues (see previous comment on colors). The colors in the legend are modified accordingly.

Fig. 14. in b) and c) put 'strong inversion' and 'weak inversion', to be consistent with the text. In k) place the compression direction arrow.

- **Author's reply:** there is some confusion here, we refer to "strong" fault as a fault that does not weaken the rocks in which it sits sufficiently to be reactivated, in contrast to a "weak fault" that is weakening the rock sufficiently for reactivation. **We have now specified that the internal friction angle of the fault is high or low, avoiding confusion.**
- Thanks for spotting the missing arrow in (k), it is now added.

Fig. 15. The words in the figure are not well-read. Improve resolution.

- **Author's reply (to the same comment by reviewer 1):** The figure in the submitted manuscript is (close) to the original resolution of the figure, as published in Jara et al. (2015). We have now inserted the highest possible resolution figure, and where needed replaced the less readable text and annotation with high-resolution text/annotation

**Citations and references:**

Cross-check citations and references.

Carefully review each reference.

- **Author's reply:** We have done a thorough check of all references, as suggested, and have corrected any errors we found. We will however do a thorough final check before resubmission.

Not all citations are in the reference list: Example: Tari et al., 2021; Kiss et al., 2020. Review.

'et al' is sometimes in normal font, others in italics. Check.

- **Author's reply:** We have added the Tari et al. (2021) and Kiss et al. (2020) details to the reference list.
- We have checked the text for instances of "et al." in italics, and have modified these where needed.

Lamplugh 1991 is cited in the text but in the references, it appears as 1919.

- **Author's reply:** This is a typo, it should be 1919 and is now corrected

Koopman 1987 is not in the reference list; I found Koopman et al 1987. Review

- **Author's reply:** Thanks for noticing, it should be Koopman et al. 1987 in the text, we have corrected it.

Cadell 1888 is cited in the text but is not in the list.

- **Author's reply:** Thanks for noticing, we have added the reference to the list

McClay is cited in the text but is not listed.

- **Author's reply:** We have gone over the references and corrected things where needed.

Jager and Geluk reference is 2007? L. 1532

- **Author's reply:** Indeed, that should be the 2007 publication, we have added the year of publication in the list, plus a link

Reference Dooley and Hudec is 2020 or 2021?

- **Author's reply:** The year of publication was 2020, it is corrected wherever needed.

Reference Pinto et al., was published in 2010, not in 2016.

- **Author's reply:** Thanks for noticing, we have corrected the error.